# Where Detectors Fail: Probing Generative Space for Generalizable AI-Generated Image Detection

Zijie Cao [1]   Weijie Tu [2]   Yao Xiao [1]   Weijian Deng [3]   Weiyan Chen [1]   Liang Lin [1 4]   Pengxu Wei [1 4]

## Abstract

Detecting AI-generated images (AIGI) remains challenging because detectors often fail to generalize to unseen generators. Although existing methods are trained on large datasets, their performance still degrades when generation settings change, indicating that data scale alone is insufficient and that limited coverage of generative variations during training is a key factor. Studies on generative model editing show that small changes in internal representations can produce diverse and meaningful image variations, many of which are not explored under standard sampling. Leveraging this insight, we propose PROBE (Probing Robustness via Boundary Exploration), a framework that improves detector generalization by actively exploring challenging regions of the generative process. Instead of treating the generator as a fixed data source, PROBE uses the detector as a critic to steer the generator through manifold-level modifications, producing realistic samples that are difficult to classify. These samples expose failure cases that are uncommon under standard data sampling strategies and are used to refine the detector. Experimental results across multiple benchmarks indicate that PROBE enhances generalization to unseen generators, resulting in more generalizable AIGI detection performance. Code and models are available at https://github.com/Amamiya-C/PROBE-AIGI-Detection.

## 1. Introduction

Modern generative models, particularly diffusion-based

[1] School of Computer Science and Engineering, Sun Yat-sen University, Guangzhou, China [2] Australian National University, Canberra, Australia [3] Tsinghua Shenzhen International Graduate School, Tsinghua University, Shenzhen, China [4] Peng Cheng Laboratory, Shenzhen, China. Correspondence to: Pengxu Wei <weipx3@mail.sysu.edu.cn>.

*Proceedings of the 43rd International Conference on Machine Learning, Seoul, South Korea. PMLR 306, 2026. Copyright 2026 by the author(s).*

models, have achieved remarkable success in producing highly realistic images (Podell et al., 2024; Rombach et al., 2022). Alongside this progress, there has been growing interest in building reliable detectors that can distinguish AI-generated images from real ones. While existing detectors often perform well on images generated by known generators, their performance degrades significantly when evaluated on images from unseen generators (Wang et al., 2020; Zhu et al., 2023b; Ojha et al., 2023; Yan et al., 2025a). This lack of generalization remains a central challenge in AI-generated image detection.

This work identifies that a key reason for this limitation lies not in model capacity or the amount of training data, but in how training data is distributed. Diffusion models do not produce images from a single rigid generation pattern. Instead, their outputs are governed by latent variables, attention mechanisms, and internal representations that jointly shape the generation dynamics (Rombach et al., 2022; Hertz et al.; Chefer et al., 2023; Mokady et al., 2023). Prior work on diffusion model editing has shown that small perturbations to these internal representations can induce realistic and semantically meaningful variations in generated images, such as changes in object attributes, appearance, or scene composition (Hertz et al.; Kadosh et al., 2025; Kawar et al., 2023; Chefer et al., 2023; Mokady et al., 2023; Gu et al., 2023). This suggests that a single trained diffusion model implicitly encodes a rich space of plausible image variations. In practice, however, standard sampling procedures explore only a limited portion of this generative space. Images are typically produced using fixed prompts and default settings, without deliberately probing challenging or ambiguous regions of the generation process. As a result, even large-scale synthetic datasets often cover only a narrow subset of the variations that the generator can produce. This mismatch helps explain why detectors trained on such data tend to overfit to generator-specific artifacts and struggle to generalize to unseen or modified generators.

A common strategy to address this issue is to collect data from multiple generators (Baraldi et al., 2024; Yan et al., 2025a; Park & Owens, 2025). While this increases diversity to some extent, it still samples only a small number of discrete points in a much larger generative space. Moreover,

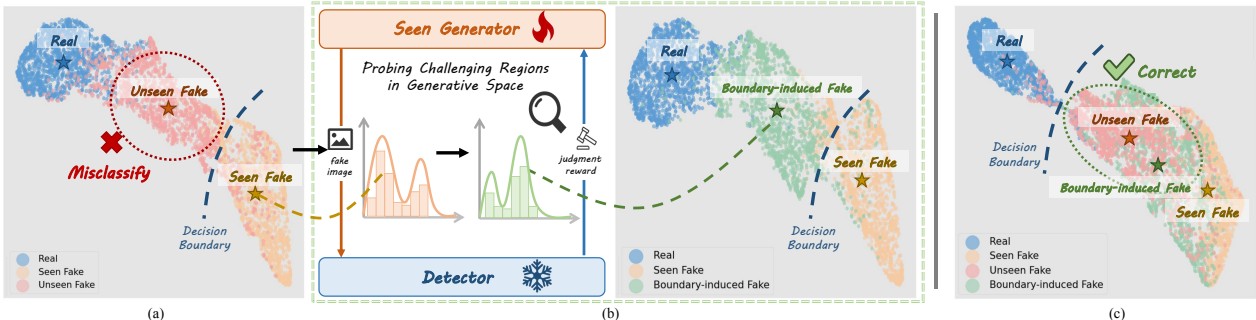

*Figure 1.* **PROBE: Improving detector generalization via boundary-induced fake samples.** (a) A detector trained on real images and samples from seen generators often fails to generalize to unseen generators, leading to misclassification of unseen fake images. (b) PROBE uses the detector as a critic to guide a seen generator toward challenging regions of the generative space. By steering generation based on detector feedback, PROBE produces boundary-induced fake samples that expose failure modes of the detector while remaining visually realistic. (c) Fine-tuning the detector on these boundary-induced fake samples reshapes the decision boundary, improving robustness and enabling correct classification of previously unseen fake images.

the rapid evolution of generative models makes continuous data collection increasingly expensive and difficult to maintain. These observations suggest that improving generalization requires not only more data, but more effective exploration of the generative space itself.

Motivated by this view, we propose PROBE, a framework for Probing Robustness via Boundary Exploration. Rather than treating the generator as a fixed source of training data, PROBE views it as a controllable mechanism for exploring the generative space. Instead of passively collecting samples, PROBE actively probes the generator to uncover regions that lie near the detector's decision boundary. These regions correspond to challenging but realistic variations that are unlikely to be observed through standard sampling.

Concretely, PROBE treats the detector as a critic and uses it to guide the generator toward regions of the generative space where the detector is most likely to fail, as illustrated in Figure 1. By optimizing the generator to produce images that are difficult for the detector to classify, PROBE exposes failure modes that are rarely encountered under standard sampling. To ensure that these samples remain realistic, we impose perceptual regularization during optimization, which prevents degenerate solutions and keeps the generated images within the natural image manifold. The resulting framework operates in two stages, as shown in Figure 1. First, the generator is optimized to explore challenging regions of the generative space, producing boundary-induced fake samples. Second, the detector is fine-tuned on these samples, allowing it to reduce reliance on generator-specific artifacts and improve generalization to unseen generators.

To validate the effectiveness of PROBE, we conduct extensive experiments across seven benchmarks, including both in-house and in-the-wild datasets that capture diverse real-world generative conditions. We compare PROBE with

a range of existing detection methods that employ different detector backbones and training strategies. The results show that PROBE consistently improves the generalization performance of baseline detectors, outperforming a recent state-of-the-art method by an average of 4.6%. Overall, our main contributions are: **1) A boundary-driven perspective on detector generalization.** We argue that limitations in AI-generated image detection stem from insufficient coverage of the generative space. Rather than data scale alone, exposing detectors to challenging variations near their decision boundaries is crucial; **2) PROBE: detector-guided boundary exploration for improved generalization.** We propose PROBE, a framework that uses the detector as a critic to guide a controllable generator toward challenging regions of the generative space. By incorporating realistic yet difficult samples into training, PROBE effectively improves generalization to unseen generators.

## 2. Related Work

**Low-level artifact-based detection methods.** This line of work focuses on detecting low-level artifacts in generated images. CNNSpot (Wang et al., 2020) introduces a one-to-many generalization setting to learn discriminative CNN features, while frequency-domain methods (Yan et al., 2025a; Corvi et al., 2023; Frank et al., 2020) exploit statistical discrepancies and spatial-domain approaches capture pixel- and texture-based cues (Liu et al., 2020; Ju et al., 2022; Zhong et al., 2024). Additionally, LGrad (Tan et al., 2023) and NPR (Tan et al., 2024) target gradient-based features and upsampling-related artifacts, respectively.

**Leveraging pre-trained vision–language models.** To improve generalization, UnivFD (Ojha et al., 2023) leverages a frozen CLIP (Radford et al., 2021) backbone with a trainable classifier, while Effort (Yan et al., 2025b) mitigates feature

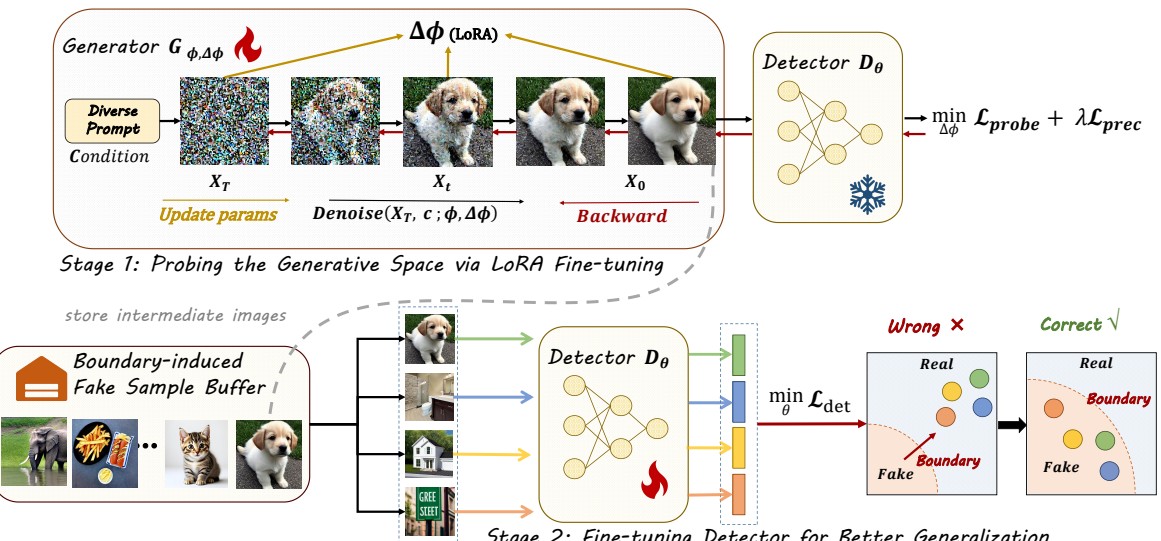

*Figure 2.* **Overall framework of PROBE.** PROBE consists of two stages: 1) Generative Space Probing: Treat the detector as a critic, use its outputs to directly supervise generator fine-tuning, probe the generative space and obtain challenging samples that reflect the detector's failure modes; 2) Detector Fine-tuning: Fine-tune the detector with the boundary-induced fake samples generated in Stage 1, which refines the decision boundary by forcing the detector to learn more robust and discriminative features.

collapse via an SVD-based strategy. FatFormer (Liu et al., 2024) and C2P-CLIP (Tan et al., 2025) further enhance CLIP-based detectors using frequency-domain features and real–fake concept injection, respectively, and RINE (Koutlis & Papadopoulos, 2024) exploits intermediate CLIP features for finer-grained discrimination.

**Improving training data of detector.** Prior work shows that detectors trained on low-quality or misaligned data often overfit generator-specific artifacts or dataset biases, motivating data-centric strategies (Ojha et al., 2023; Grommelt et al., 2024; Guillaro et al., 2025). Representative approaches include data augmentation (Li et al., 2025; Wang et al., 2020), alignment via compression and resizing (Grommelt et al., 2024), large-scale multi-generator datasets (Park & Owens, 2025; Baraldi et al., 2024), generator-based synthesis of challenging samples through reconstruction or semantic alignment (Cazenavette et al., 2024; Chen et al., 2024; Rajan et al., 2025; Guillaro et al., 2025; Chen et al.) and feature-space adversarial hard-sample synthesis via teacher-student discrepancy maximization (Zhu et al., 2023a).

**Diffusion model editing.** The generative process is a dynamic evolution shaped by attention mechanisms, latent variables, and internal representations. Building on this, a line of work has explored efficient diffusion model editing, such as modulating cross-attention to control spatial layout and semantic fidelity (Hertz et al.; Chefer et al., 2023), optimizing text embeddings for high-fidelity image editing (Kawar et al., 2023; Mokady et al., 2023), and fine-

tuning model weights via Low-Rank Adaptation (LoRA) for multi-concept customization (Hu et al., 2022; Gu et al., 2023). Collectively, these studies show that subtle perturbations to internal generative dynamics can yield realistic and semantically meaningful image variations.

**Using detector feedback for generation.** DADA (Zhou et al., 2025) injects adversarial perturbations into the denoising trajectory to synthesize hard false positives for polyp detection, without modifying the generator's parameters. RealGen (Ye et al., 2025) optimizes the generator with detector-based rewards to improve photorealism. These methods target a specific detection task or generation quality, rather than cross-generator generalization of AIGI detectors. In contrast to prior data-centric detection methods that operate in image space or rely on standard sampling and reconstruction, and to adversarial augmentation methods that act locally in input or feature space without realism constraints, our approach leverages insights from generative model editing to perturb internal generator representations, performing realism-constrained, boundary-aware exploration of the generative space. This yields diverse and challenging samples that lie beyond standard sampling, exposing failure modes unreachable by input-space or feature-space alternatives.

## 3. PROBE: Probing Robustness via Boundary Exploration

**Overview.** We begin by illustrating the core challenge addressed in this work. As shown in Figure 1(a), a detector

*Prompt: "A cool dog with grey and white fur wearing a collar."*

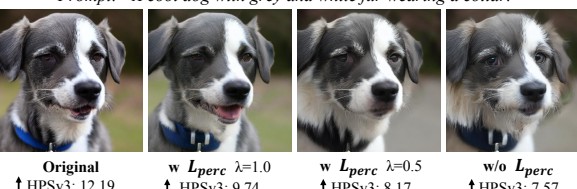

| **Original** | **w $L_{perc}$** $\lambda$=1.0 | **w $L_{perc}$** $\lambda$=0.5 | **w/o $L_{perc}$** |
|---|---|---|---|
| ↑ HPSv3: 12.19 | ↑ HPSv3: 9.74 | ↑ HPSv3: 8.17 | ↑ HPSv3: 7.57 |

*Figure 3.* **Effect of perceptual regularization.** We measure image quality and text–image alignment using HPSv3 (Ma et al., 2025). Removing perceptual regularization leads to degraded visual fidelity and semantic inconsistency, while moderate regularization preserves realism and stabilizes generation.

trained on real images and samples from seen generators often learns a decision boundary that performs well on familiar data but fails to generalize to images produced by unseen generators. These unseen fake images fall outside the training distribution, leading to systematic misclassification. This reveals a key limitation of existing training paradigms: the training data covers only a limited portion of the underlying generative space.

To address this issue, PROBE explicitly targets regions of the generative space that are most informative for improving generalization. As illustrated at the top of Figure 2, instead of treating the generator as a fixed data source, we use the detector itself as a critic to guide the generator toward challenging regions of the generative space. This produces boundary-induced fake samples that expose failure modes of the detector while remaining visually realistic. These samples are illustrated in the feature space in Figure 1(b).

Finally, as shown in Figure 2 and Figure 1(c), fine-tuning the detector on these boundary-induced samples reshapes the decision boundary, enabling correct classification of previously unseen fake images. This allows the detector to move beyond reliance on generator-specific artifacts and achieve improved robustness under distribution shifts.

### 3.1. Detector-Guided Generative Boundary Exploration

Let $G_\phi$ denote a pretrained diffusion generator and $D_\theta$ a detector that predicts the probability of an image being fake. Given a noise vector $\mathbf{x}_T \sim \mathcal{N}(0, I)$ and a conditioning input $\mathbf{c}$ (*e.g.*, a text prompt), the generator produces an image $\mathbf{x} = G_\phi(\mathbf{x}_T, \mathbf{c})$. To explore challenging regions of the generative space in a controlled manner, we use lightweight trainable parameters using Low-Rank Adaptation (LoRA) (Hu et al., 2022). Specifically, LoRA modules are applied to the attention layers of the diffusion U-Net (Qin et al., 2020), yielding an adapted generator $\mathbf{x} = G_{\phi,\Delta\phi}(\mathbf{x}_T, \mathbf{c})$, where the original parameters $\phi$ are frozen and only the low-rank parameters $\Delta\phi$ are optimized.

PROBE steers the generator using feedback from the detector. Intuitively, we encourage the generator to produce

images that are likely to be misclassified as real by the detector, thereby exposing decision-boundary regions. Formally, the detector outputs a probability $D_\theta(\mathbf{x}) = p_\theta(\text{fake} \mid \mathbf{x})$. We optimize the generator by minimizing

$$\mathcal{L}_{\text{probe}} = \mathbb{E}_{\mathbf{x}_T, \mathbf{c}} \left[ -\log(1 - p_\theta(\text{fake} \mid G_{\phi,\Delta\phi}(\mathbf{x}_T, \mathbf{c}))) \right].$$

This objective encourages the generator to explore regions of the generative space that are insufficiently captured by the detector, yielding realistic samples that expose its generalization weaknesses.

**Efficient optimization via deep reward tuning.** Directly backpropagating through the full diffusion trajectory is computationally expensive, as it requires storing and differentiating through all denoising steps. To enable efficient optimization, we adopt Deep Reward Tuning (Wu et al., 2024), which allows gradients defined on the final generated image to guide the generation process without backpropagating through every diffusion step.

Combined with LoRA, this strategy enables efficient and stable optimization while preserving the original generative behavior of the diffusion model.

**Perceptual regularization.** Optimizing the generator solely based on detector feedback may lead to degenerate or visually implausible outputs. To ensure realism, we use the perceptual regularization (Johnson et al., 2016):

$$\mathcal{L}_{\text{perc}} = \mathbb{E}_{\mathbf{x}_T, \mathbf{c}} \left[ \|\gamma(G_{\phi,\Delta\phi}(\mathbf{x}_T, \mathbf{c})) - \gamma(G_\phi(\mathbf{x}_T, \mathbf{c}))\|_2^2 \right],$$

where $\gamma(\cdot)$ denotes a VGG-19 (Simonyan & Zisserman, 2014)] based fixed perceptual feature extractor. As illustrated in Fig. 3, perceptual regularization plays a crucial role in maintaining visual fidelity during generator optimization.

The final generator optimization objective is

$$\min_{\Delta\phi} \quad \mathcal{L}_{\text{probe}} + \lambda\mathcal{L}_{\text{perc}}, \tag{1}$$

where $\lambda$ controls the weight of $\mathcal{L}_{\text{perc}}$. This objective ensures that generated samples remain visually realistic while still exposing challenging failure cases.

### 3.2. Boundary-Aware Detector Adaptation

After boundary exploration, we fine-tune the detector $D_\theta$ using a mixture of real images, standard diffusion-generated images, and boundary-induced fake samples. Let $\mathcal{D}_{\text{real}} = \{(\mathbf{x}_i, y_i = 0)\}_{i=1}^{N_r}$, $\mathcal{D}_{\text{gen}} = \{(\mathbf{x}_j, y_j = 1)\}_{j=1}^{N_g}$, $\mathcal{D}_{\text{probe}} = \{(\mathbf{x}_k, y_k = 1)\}_{k=1}^{N_p}$, denote the real images, standard generated images, and boundary-induced samples, respectively. Full training data is $\mathcal{D} = \mathcal{D}_{\text{real}} \cup \mathcal{D}_{\text{gen}} \cup \mathcal{D}_{\text{probe}}$. The detector minimization objective is

$$\min_\theta \ \mathcal{L}_{\text{det}} = \mathbb{E}_{(\mathbf{x},y)\sim\mathcal{D}} \left[ \mathcal{L}_{\text{BCE}}(D_\theta(\mathbf{x}), y) \right]. \tag{2}$$

By exposing regions of the generative space that are not well represented in standard sampling, the boundary-induced samples enable the detector to learn more robust decision rules and improve generalization to unseen generators.

The boundary exploration and detector adaptation process could be applied iteratively by repeatedly updating the generator using the refined detector. However, we observe that a single round is sufficient in practice. On ResNet-50 detector, the first round improves the average bAcc by 14.6%, while two further rounds add 1.3% and 0.7%. The initial exploration already exposes the dominant regions of the generative space that are underrepresented in standard sampling and most responsible for detector failures. Further iterations tend to revisit similar variations, yielding marginal gains while incurring substantially higher computational cost. We therefore adopt a single-round strategy, which achieves a favorable balance between performance and efficiency. Full per-iteration results are in the Appendix G.

## 4. Experiments

### 4.1. Experimental Setup

**Datasets.** We evaluate detectors on both in-house and in-the-wild datasets. In-house datasets include GenImage (Zhu et al., 2023b), Synthbuster (Bammey, 2023), and AIGI-Quality-Paradox (Xiao et al., 2025). Notably, GenImage exhibits compression bias (Grommelt et al., 2024); following established evaluation protocols (Chen et al.; Guillaro et al., 2025), we apply uniform JPEG compression with a quality factor of 96 to all generated images to mitigate its impact. AIGI-Quality-Paradox further features state-of-the-art generators, diverse real image sources, and semantic alignment between real and fake image pairs. In-the-wild datasets: Chameleon (Yan et al., 2025a), WildRF (Cavia et al., 2024), SynthWildX (Cozzolino et al., 2024), and AIGI-Bench (Li et al., 2026), comprise images collected from social media and AI art communities that exhibit high content diversity and platform-induced artifacts such as compression and blurring, thereby better reflecting real-world usage scenarios.

**Evaluation metrics.** Following prior studies (Wang et al., 2020; Zhu et al., 2023b; Tan et al., 2024; Ojha et al., 2023; Chen et al., 2024), we adopt balanced accuracy (bAcc) as the evaluation metric, using a classification threshold of 0.5. Balanced accuracy is defined as the average of accuracies on fake and real samples. In addition, we report the average precision scores in the Appendix.

**Compared methods.** To provide a comprehensive comparison, we evaluate eight competitive detectors, including six methods trained on GenImage SD 1.4 split: CNNSpot (Wang et al., 2020), UnivFD (Ojha et al., 2023),

DRCT (Chen et al., 2024)[1], NPR (Tan et al., 2024), Effort (Yan et al., 2025b), and AIDE (Yan et al., 2025a). In addition, we include two methods trained on datasets constructed by reconstructing images from MSCOCO (Lin et al., 2014) using Stable Diffusion 2.1 (Reconstruction Training Set): BFree (Guillaro et al., 2025) and DDA (Chen et al.). Note that for CNNSpot and UnivFD, we retrain the models on the GenImage SD 1.4 (Zhu et al., 2023b) training set, as this consistently yields better average performance across benchmarks. For the remaining methods, we evaluate them using their officially released checkpoints.

**Implementation details of PROBE.** We train two models with different detector backbones on two training sets, respectively. On GenImage SD 1.4, we use ResNet-50 (He et al., 2016) as the detector and Stable Diffusion 1.4 as the generator. On the reconstruction training set, we adopt a pretrained DINOv2-ViT-L (Oquab et al., 2024) as the detector and use Stable Diffusion 2.1 as the generator. The generator in each setting is used to construct the corresponding detector training set and is further fine-tuned by PROBE using the first 20k prompts from COCO2014 (Lin et al., 2014). More implementation details are in the Appendix F.

### 4.2. Main Results

**PROBE consistently improves detection performance over baseline counterparts.** Table 1 presents performance comparisons across seven benchmarks. We observe that PROBE improves detection accuracy for both detector backbones, achieving an average balanced accuracy gain of 14.6% with ResNet-50 and 6.5% with DINOv2 across both in-house and in-the-wild datasets. These improvements hold for both CNN-based and transformer-based backbones, demonstrating effectiveness across architectures.

**PROBE achieves higher detection accuracy than compared AIGI detectors.** As shown in Table 1, among methods trained on GenImage SD 1.4, PROBE-ResNet50 achieves higher or comparable performance relative to the compared approaches. Particularly, it outperforms the respective second-best methods by more than 10% on AIGI-Quality-Paradox and Chameleon, and exceeds the second-best average accuracy across all seven benchmarks by 7.2%. Unlike UnivFD, Effort, and AIDE, which leverage knowledge from pretrained vision–language models, PROBE-ResNet50 attains strong performance without relying on such models, suggesting that PROBE effectively adapts ResNet's decision boundary by exploiting generative boundaries of Stable Diffusion 1.4, leading to improved accuracy.

Moreover, PROBE-DINOv2 achieves the highest perfor-

---

[1]We use DRCT-ConvB owing to its better average performance across benchmarks compared to other DRCT variants.

*Table 1.* **Overall comparison across seven benchmarks.** The evaluation metric is Balanced Accuracy (bAcc%). Methods are grouped into two categories based on the training data: the GenImage SD 1.4 split and datasets constructed by reconstructing real images using SD 2.1 (Reconstruction Training Set). For each group, the best result is marked in **bold** and the second best is underlined. We observe that PROBE-DINOv2 achieves the highest bAcc across all benchmarks, demonstrating the effectiveness of PROBE.

| Method | GenImage | AIGI-Quality-Paradox | Synthbuster | Chameleon | SynthWildX | WildRF | AIGI-Bench | Avg. |
|---|---|---|---|---|---|---|---|---|
| *Trained on GenImage SD 1.4* | | | | | | | | |
| CNNSpot (Wang et al., 2020) | 71.1 | 63.3 | 68.4 | 61.5 | 60.2 | 64.0 | 55.0 | 63.4 |
| UnivFD (Ojha et al., 2023) | 77.0 | 73.4 | **79.2** | 61.3 | 61.3 | 57.5 | 58.2 | 66.8 |
| NPR (Tan et al., 2024) | **80.0** | 63.7 | 54.1 | 56.4 | 56.0 | 63.8 | 59.4 | 61.9 |
| DRCT (Chen et al., 2024) | 77.0 | 72.0 | 71.3 | 63.9 | 70.7 | 74.1 | 67.4 | 70.9 |
| AIDE (Yan et al., 2025a) | 61.2 | 67.7 | 56.2 | 63.1 | 56.9 | 60.5 | 61.1 | 61.0 |
| Effort (Yan et al., 2025b) | 79.6 | 72.9 | 75.7 | 60.3 | 58.5 | 72.1 | 59.8 | 68.4 |
| ResNet50 | 72.6 | 64.8 | 65.4 | 58.6 | 60.1 | 66.8 | 56.4 | 63.5 |
| **PROBE (ResNet50)** | 78.4 (+5.8%) | **87.7 (+22.9%)** | 79.1 (+13.7%) | **74.0 (+15.4%)** | **75.6 (+15.5%)** | **81.3 (+14.5%)** | **70.4 (+14.0%)** | **78.1 (+14.6%)** |
| *Trained on Reconstruction Training Set* | | | | | | | | |
| BFree (Guillaro et al., 2025) | 89.2 | 91.9 | 95.2 | 76.0 | 95.7 | 93.3 | 83.5 | 89.3 |
| DDA (Chen et al.) | 91.7 | 90.1 | 89.9 | 82.4 | 90.9 | 90.3 | 81.6 | 88.1 |
| DINOv2 | 86.8 | 87.9 | 91.9 | 77.0 | 93.5 | 92.9 | 81.7 | 87.4 |
| **PROBE (DINOv2)** | **96.8 (+10.0%)** | **92.9 (+5.0%)** | **97.5 (+5.6%)** | **86.6 (+9.6%)** | **96.4 (+2.9%)** | **96.9 (+4.0%)** | **89.9 (+8.2%)** | **93.9 (+6.5%)** |

*Table 2.* **Comparison of GenImage (Zhu et al., 2023b) and AIGI-Quality-Paradox (Xiao et al., 2025) using balanced accuracy (%).** We observe that PROBE-DINOv2 achieves the highest average bAcc on both datasets, suggesting its strong and consistent effectiveness.

| Method | GenImage | | | | | | | | | AIGI-Quality-Paradox | | | | | | |
|---|---|---|---|---|---|---|---|---|---|---|---|---|---|---|---|---|
| | Midjourney | SD 1.4 | SD 1.5 | ADM | GLIDE | Wukong | VQDM | BigGAN | Avg. | FLUX | Infinity | PixArt-α | SD XL | SD 2.1 | SD 3 | Avg. |
| *Trained on GenImage SD 1.4* | | | | | | | | | | | | | | | | |
| CNNSpot (Wang et al., 2020) | 63.0 | 99.2 | 99.0 | 50.6 | 55.9 | 98.2 | 51.0 | 52.0 | 71.1 | 55.6 | 61.3 | 67.6 | 62.4 | 71.5 | 61.3 | 63.3 |
| UnivFD (Ojha et al., 2023) | 80.0 | 97.0 | 96.9 | 55.3 | 72.2 | 92.8 | 60.4 | 61.6 | 77.0 | 72.6 | 76.2 | 76.2 | 67.2 | 75.5 | 72.5 | 73.4 |
| NPR (Tan et al., 2024) | 79.9 | 86.3 | 86.3 | **76.9** | 89.9 | 87.0 | 71.7 | 61.8 | **80.0** | 62.4 | 69.4 | 63.9 | 55.8 | 68.1 | 62.8 | 63.7 |
| DRCT (Chen et al., 2024) | 87.7 | 99.9 | 99.8 | 54.3 | 55.7 | 99.8 | 64.2 | 54.7 | 77.0 | 54.8 | 83.2 | 83.1 | 79.3 | 67.0 | 64.7 | 72.0 |
| AIDE (Yan et al., 2025a) | 60.2 | 78.8 | 72.1 | 50.4 | 54.7 | 72.1 | 50.8 | 50.7 | 61.2 | 56.6 | 62.1 | 65.8 | 54.0 | 53.4 | 54.2 | 67.7 |
| Effort (Yan et al., 2025b) | 76.4 | 96.0 | 95.8 | 69.8 | 83.4 | 90.2 | **73.2** | 51.6 | 79.6 | 68.6 | 70.3 | 77.5 | 72.6 | 74.6 | 73.9 | 72.9 |
| ResNet50 | 72.7 | 99.8 | 99.7 | 51.0 | 53.6 | 97.0 | 57.4 | 50.0 | 72.6 | 55.0 | 55.2 | 77.6 | 62.2 | 74.2 | 64.7 | 64.8 |
| **PROBE (ResNet50)** | **90.7** | 99.4 | 99.2 | 60.0 | 63.0 | 99.4 | 66.9 | 49.4 | 78.4 (+5.8%) | **76.8** | **88.0** | **91.2** | **93.9** | **93.3** | 82.9 | **87.7 (+22.9%)** |
| *Trained on Reconstruction Training Set* | | | | | | | | | | | | | | | | |
| BFree (Guillaro et al., 2025) | 95.2 | 98.8 | 98.8 | 78.4 | 85.9 | 98.8 | 89.2 | 68.8 | 89.2 | 65.3 | 97.9 | 97.9 | 97.9 | 97.8 | 94.7 | 91.9 |
| DDA (Chen et al.) | 95.6 | 98.7 | 98.7 | 89.5 | 89.6 | 98.7 | 76.5 | 86.5 | 91.7 | 78.1 | 93.4 | 92.7 | 93.6 | 93.3 | 88.4 | 89.9 |
| DINOv2 | 84.0 | 99.8 | 99.7 | 72.7 | 72.8 | 99.7 | 85.6 | 80.2 | 86.8 | 49.3 | 95.4 | 95.4 | **99.7** | **99.6** | 88.1 | 87.9 |
| **PROBE (DINOv2)** | **98.8** | 99.4 | 99.4 | **91.0** | **92.1** | **99.5** | **98.9** | **95.0** | **96.8 (+10.0%)** | 66.8 | **98.3** | **98.3** | 98.3 | 98.3 | **97.5** | **92.9 (+5.0%)** |

mance among all considered approaches across seven benchmarks. This result indicates that, when combined with representations from foundation models, PROBE enables even stronger AIGI detection performance. As shown in Table 2, PROBE-DINOv2 consistently achieves strong AIGI detection performance across both GenImage and AIGI-Quality-Paradox, with average bAcc improvements of 5.1% and 1.0% over the second-best method, respectively. In addition, as shown in Tables 3 and 4, PROBE-DINOv2 generalizes well to images produced by other diffusion models. This suggests that PROBE captures shared generative artifacts across diverse models, and that exploiting generator-side variations improves cross-generator generalization. The strong results of PROBE-DINOv2 on both in-house and in-the-wild datasets also demonstrate its effectiveness and applicability in diverse and complex real-world scenarios.

**PROBE generalizes beyond diffusion-based generators.** To further investigate whether PROBE transfers to architecturally distinct generator families unseen during boundary exploration, we evaluate PROBE-DINOv2 on GAN subsets from AIGCDetectBenchmark (Zhong et al., 2024) and autoregressive samples (NOVA) from EvalGEN (Chen et al.). As shown in Table 5, PROBE-DINOv2 achieves 93.8% av-

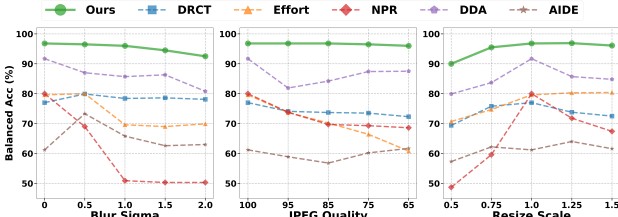

*Figure 4.* **Robustness analysis of PROBE-DINOv2 and comparison methods on GenImage under common image postprocessing operations.** PROBE-DINOv2 demonstrates strong robustness and consistently outperforms the compared methods.

erage balanced accuracy, outperforming both the DINOv2 baseline (+10.1%) and BFree (90.1%). This indicates that probing a single diffusion model exposes detector weaknesses shared across generator families, enabling generalization well beyond the probing generator itself.

## 5. Discussion

**Impact of detector backbone.** We investigate the influence of the detector backbone on the effectiveness of PROBE. Following the training setup of PROBE-DINOv2 described in Section 4.1, we replace the detector backbone with the vi-

*Table 3.* **Comparison on Synthbuster (Bammey, 2023) using balanced accuracy (%).** PROBE-DINOv2 achieves the best overall performance, suggesting strong generalization to diverse unseen generators.

| Method | DALL·E 2 | DALL·E 3 | Firefly | GLIDE | Midjourney | SD 1.3 | SD 1.4 | SD 2 | SD XL | Avg. |
|---|---|---|---|---|---|---|---|---|---|---|
| *Trained on GenImage SD 1.4* | | | | | | | | | | |
| CNNSpot (Wang et al., 2020) | 56.5 | 64.0 | 54.0 | 56.4 | 56.7 | **97.2** | **97.0** | 66.6 | 66.8 | 68.4 |
| UnivFD (Ojha et al., 2023) | 67.2 | **93.1** | **75.9** | 75.4 | 72.0 | 91.1 | 91.3 | 73.4 | 74.2 | **79.2** |
| NPR (Tan et al., 2024) | 51.6 | 46.6 | 54.7 | 55.6 | 55.8 | 57.6 | 57.5 | 53.8 | 53.5 | 54.1 |
| DRCT(ConvB) (Chen et al., 2024) | 44.8 | 49.4 | 46.6 | 52.8 | 89.8 | 92.1 | 92.1 | 89.9 | 84.1 | 71.3 |
| AIDE (Yan et al., 2025a) | 39.2 | 39.0 | 25.7 | 65.9 | 59.7 | 75.1 | 74.9 | 55.9 | 70.7 | 56.2 |
| Effort (Yan et al., 2025b) | **67.9** | 79.3 | 56.1 | **79.0** | 77.2 | 81.6 | 81.7 | 77.9 | 80.5 | 75.7 |
| ResNet50 | 49.0 | 54.5 | 49.0 | 52.0 | 65.2 | 96.9 | 96.3 | 66.1 | 59.7 | 65.4 |
| **PROBE (ResNet50)** | 48.3 | 90.2 | 46.9 | 57.4 | **93.2** | 94.9 | 94.9 | **93.1** | **93.0** | 79.1 (+13.7%) |
| *Trained on Reconstruction Training Set* | | | | | | | | | | |
| BFree (Guillaro et al., 2025) | 93.1 | 95.7 | 97.2 | 85.0 | 97.1 | 97.3 | 97.2 | 97.2 | 97.3 | 95.2 |
| DDA (Chen et al.) | 86.3 | 90.0 | 91.9 | 76.5 | 93.5 | 92.9 | 92.7 | 93.3 | 93.5 | 90.1 |
| DINOv2 | 82.0 | 88.6 | 96.3 | 65.0 | 96.9 | **99.6** | **99.9** | **99.8** | **99.7** | 91.9 |
| **PROBE (DINOv2)** | **94.6** | **99.4** | **99.5** | **86.7** | **99.5** | 99.5 | 99.5 | 99.5 | 99.5 | **97.5** (+5.6%) |

*Table 4.* **Comparison on four in-the-wild benchmarks using balanced accuracy (%)**, including Chameleon (Yan et al., 2025a), SynthWildX (Cozzolino et al., 2024), WildRF (Cavia et al., 2024), and AIGI-Bench (Li et al., 2026). PROBE consistently improves the baseline detector, and PROBE-DINOv2 achieves the best performance across four benchmarks.

| Method | Chameleon | SynthWildX | | | | WildRF | | | | AIGI-Bench | | |
|---|---|---|---|---|---|---|---|---|---|---|---|---|
| | | DALL·E 3 | Firefly | Midjourney | Avg. | Facebook | Reddit | Twitter | Avg. | CommunityAI | SocialRF | Avg. |
| *Trained on GenImage SD 1.4* | | | | | | | | | | | | |
| CNNSpot (Wang et al., 2020) | 61.5 | 64.9 | 56.7 | 59.0 | 64.9 | 59.7 | 65.0 | 67.4 | 64.0 | 53.1 | 56.8 | 55.0 |
| UnivFD (Ojha et al., 2023) | 61.3 | 65.9 | 60.7 | 57.5 | 61.4 | 58.3 | 59.3 | 54.9 | 57.5 | 60.1 | 56.3 | 58.2 |
| NPR (Tan et al., 2024) | 56.4 | 55.4 | 55.1 | 57.4 | 56.0 | 61.9 | 70.1 | 59.3 | 63.8 | 59.3 | 59.5 | 59.4 |
| DRCT(ConvB) (Chen et al., 2024) | 63.9 | 83.9 | 53.8 | 74.3 | 70.7 | 76.9 | 71.5 | 73.7 | 74.1 | 67.0 | 67.8 | 67.4 |
| AIDE (Yan et al., 2025a) | 63.1 | 67.6 | 50.3 | 52.9 | 56.9 | 59.4 | 71.9 | 50.1 | 60.5 | 61.8 | 60.5 | 61.1 |
| Effort (Yan et al., 2025b) | 60.3 | 59.4 | **60.9** | 55.3 | 58.5 | 74.1 | **79.3** | 63.0 | 72.1 | 58.6 | 61.1 | 59.8 |
| ResNet50 | 58.6 | 72.0 | 50.2 | 58.1 | 60.1 | 66.6 | 67.1 | 66.7 | 66.8 | 51.7 | 61.0 | 56.4 |
| **PROBE (ResNet50)** | **74.0** (+15.4%) | **90.3** | 56.7 | **79.7** | **75.6** (+15.5%) | **87.8** | 72.9 | **83.3** | **81.3** (+14.5%) | **71.0** | **69.9** | **70.4** (+14.0%) |
| *Trained on Reconstruction Training Set* | | | | | | | | | | | | |
| BFree (Guillaro et al., 2025) | 76.0 | 95.7 | 95.7 | 95.7 | 95.7 | 95.6 | 86.2 | 97.3 | 93.3 | 81.7 | 85.2 | 83.5 |
| DDA (Chen et al.) | 82.4 | 92.3 | 87.3 | 93.1 | 90.9 | 93.1 | 86.4 | 91.5 | 90.3 | 86.7 | 76.4 | 81.6 |
| DINOv2 | 77.0 | 93.2 | 95.0 | 92.2 | 93.5 | 94.9 | 88.9 | 94.8 | 92.9 | 79.2 | 84.3 | 81.7 |
| **PROBE (DINOv2)** | **86.6** (+9.6%) | **96.5** | **96.4** | **96.5** | **96.4** (+2.9%) | **98.4** | **93.4** | **99.0** | **96.9** (+4.0%) | **89.8** | **89.9** | **89.9** (+8.2%) |

sual encoders of CLIP-B/16 and CLIP-L/14 (Radford et al., 2021), as well as DINOv2-B/16, and report balanced accuracy across five benchmarks. We additionally compare our results with BFree, which achieves the second-highest average accuracy across the seven benchmarks.

The results are summarized in Table 6. We observe that, relative to their corresponding baseline counterparts, PROBE consistently improves detection accuracy across all benchmarks. Moreover, when comparing backbones with similar parameter scales (*e.g.*, CLIP-B/16 *vs.* DINOv2-B/16), PROBE combined with DINOv2 generally achieves stronger detection performance. Compared to BFree, the second-best method, PROBE achieves competitive or higher balanced accuracy, except for PROBE-CLIP-B/16, demonstrating the robustness of PROBE to different backbone choices.

**Robustness against image post-processing.** To validate robustness against common image post-processing operations, we evaluate all detection methods on the GenImage-JPEG96 dataset under Gaussian blurring (sigma: 0–2), JPEG compression (quality: 95–65), and resizing (scale: 0.5–1.5). For PROBE, we report results using the DINOv2 backbone, and balanced accuracy is adopted as the evaluation metric. As shown in Figure 4, PROBE-DINOv2 exhibits strong robustness across all three post-processing techniques over a

wide range of transformations strengths. Even under high-strength post-processing, including blur sigma 2.0, JPEG quality 65, and scale 0.5, its balanced accuracy remains above 90%, outperforming the respective second-best methods by 11.7%, 8.5%, and 10.1%. In contrast, NPR, which primarily relies on low-level artifacts, suffers noticeable performance degradation under blurring and resizing, whereas PROBE maintains consistently high and stable performance, suggesting that it learns more robust and generalizable features for AIGI detection.

**Comparison with Data Augmentation Methods.** To further validate that PROBE meaningfully adapts the detector's decision boundary rather than increasing data quantity, we compare PROBE with three data augmentation techniques. Specifically, we consider: **(a)** SD 1.4 Standard Sampling (SD 1.4 standard): images generated by Stable Diffusion 1.4 using the same prompts employed for generator fine-tuning in PROBE; **(b)** Pixel-level Adversarial Augmentation: adversarial samples constructed via PGD attacks (Madry et al., 2017) applied directly in the image (pixel) space of SD 1.4 standard; **(c)** Latent-level Adversarial Augmentation: adversarial samples generated by applying PGD attacks to the latent variables of the diffusion model during generation process of SD 1.4 standard.

*Table 5.* **Comparison of more generators using balanced accuracy (%).** PROBE consistently improves the baseline on both GAN-based and autoregressive generators, suggesting strong generalization to unseen generators with different architectures.

| Method | BigGAN | GauGAN | CycleGAN | ProGAN | StyleGAN | StyleGAN2 | NOVA | Avg. |
|---|---|---|---|---|---|---|---|---|
| BFree (Guillaro et al., 2025) | 91.9 | 97.0 | **86.1** | 95.9 | 82.1 | 78.0 | 99.5 | 90.1 |
| DINOv2 | 85.1 | 96.1 | 65.4 | 93.0 | 78.1 | 74.0 | 94.2 | 83.7 |
| **PROBE (DINOv2)** | **96.0** | **99.0** | 83.4 | **97.3** | **91.0** | **90.4** | **99.8** | **93.8** (+10.1%) |

*Table 6.* **Impact of Detector Backbones.** We investigate the impact of different detector backbones, with experiments conducted on the reconstruction training set. For each column, the best result is marked in **bold** and the second best is underlined. PROBE consistently improves performance across different backbones, with DINOv2 achieving the strongest overall performance.

| Method | Gen-Image | Chame-leon | Synth-WildX | Wild-RF | AIGI-Bench | Avg. |
|---|---|---|---|---|---|---|
| BFree | 89.2 | 76.0 | 95.7 | 93.3 | 83.5 | 87.5 |
| CLIP-B/16 | 81.5 | 80.2 | 76.9 | 80.9 | 77.4 | 79.4 |
| +PROBE | 82.9 | 85.4 | 85.3 | 86.7 | 83.5 | 84.7 (+5.3%) |
| CLIP-L/14 | 88.8 | 81.2 | 92.7 | 90.7 | 83.9 | 87.5 |
| +PROBE | 91.9 | **88.6** | 93.0 | 93.2 | 87.0 | 90.7 (+3.2%) |
| DINOv2-B/16 | 85.9 | 77.8 | 91.4 | 92.4 | 82.0 | 85.9 |
| +PROBE | 91.3 | 85.1 | 95.1 | 93.4 | 87.6 | 90.5 (+4.6%) |
| DINOv2-L/14 | 86.8 | 77.0 | 93.5 | 92.9 | 81.7 | 86.4 |
| +PROBE | **96.8** | 86.6 | **96.4** | **96.9** | **89.9** | **93.3** (+6.9%) |

*Table 7.* **Comparison with three data augmentation methods.** We consider four augmentation strategies: (a) generating additional data using Stable Diffusion 1.4, (b) pixel-level adversarial augmentation, (c) latent-level adversarial augmentation, and (d) PROBE. For each column, the best result is marked in **bold** and the second best is underlined. The results show that PROBE yields larger performance gains for the ResNet-50 baseline detector than the other augmentation methods, highlighting its effectiveness.

| Augmentation Method | Gen-Image | Chame-leon | Synth-WildX | Wild-RF | AIGI-Bench | Avg. |
|---|---|---|---|---|---|---|
| ResNet50 | 72.6 | 58.6 | 60.1 | 66.8 | 56.4 | 62.9 |
| a) SD 1.4 standard | 73.7 | 64.5 | 65.5 | 73.1 | 60.5 | 67.5 (+4.6%) |
| b) Pixel-level aug | 77.1 | 70.8 | 67.9 | 71.5 | 65.7 | 70.6 (+7.7%) |
| c) Latent-level aug | 74.8 | 65.6 | 70.1 | 77.1 | 63.4 | 70.2 (+7.3%) |
| d) PROBE | **78.4** | **74.0** | **75.6** | **81.3** | **70.4** | **76.0** (+13.1%) |

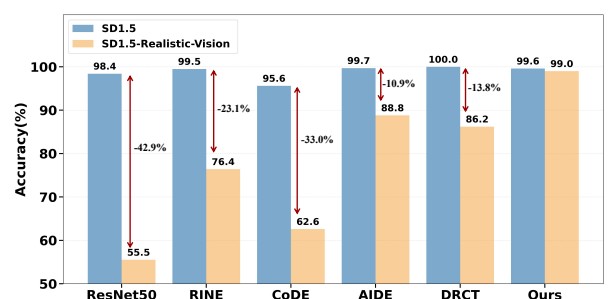

*Figure 5.* **Generalization under high-quality generation.** We evaluate multiple detectors on images generated by Stable Diffusion 1.5 and SD 1.5-Realistic-Vision. SD 1.5-Realistic-Vision is designed to produce more photorealistic images through fine-tuning on high-quality data. Both generators are unseen for all detectors during training. While most detectors perform well on SD 1.5, their performance drops substantially on SD 1.5-Realistic-Vision, despite identical generation settings (*i.e.*, seed, prompt, sampling steps, sampling method, scheduler, resolution, and CFG scale). In contrast, our method maintains strong performance.

*Table 8.* **Effect of combining boundary-induced samples from multiple detectors.** We fine-tune the ResNet50 detector using two sets of PROBE samples with identical data volume: (i) samples generated using ResNet50 as the critic, and (ii) samples aggregated from multiple detectors (ResNet50, UnivFD, and DRCT). The best result is marked in **bold** in each column.

| Method | Gen-Image | Chame-leon | Synth-WildX | Wild-RF | AIGI-Bench | Avg. |
|---|---|---|---|---|---|---|
| ResNet50 | 72.6 | 58.6 | 60.1 | 66.8 | 56.4 | 62.9 |
| + ResNet50 samples | 78.4 | **74.0** | 75.6 | 81.3 | 70.1 | 76.0 (+13.1%) |
| + Multiple Detectors samples | **79.2** | 73.5 | **76.8** | **82.7** | **71.2** | **76.7** (+13.8%) |

The experimental results in Table 7 lead to two observations. **First**, standard SD 1.4 sampling yields a moderate performance gain over the baseline, indicating that increased exposure to generated images helps mitigate overfitting. In contrast, PROBE provides substantially larger improvements, suggesting that exploring the generative boundary, rather than data quantity alone, is key to its effectiveness. **Second**, although both pixel-level and latent-level adversarial examples improve baseline performance, their gains remain consistently smaller than those achieved by PROBE, indicating that PROBE promotes robustness beyond what these augmentation strategies provide.

**Robustness under realistic generation.** To evaluate robustness under realistic generation shifts, we consider SD 1.5-Realistic-Vision (SG_161222, 2024), a widely used

model designed to generate higher-quality and more photorealistic images. As shown in Figure 5, detectors perform well on images generated by SD 1.5, but exhibit significant performance degradation when evaluated on SD 1.5-Realistic-Vision. This indicates that even improvements in visual quality alone can induce substantial distribution shifts that challenge existing detectors. In contrast, our method gives consistently high performance across both generators.

**Boundary-induced samples across different detectors.** Figure 6 visualizes boundary-induced fake samples generated using different detectors as critics separately, all projected into the feature space of the ResNet50 detector (anchor). Despite being guided by different models, these samples exhibit substantial overlap, indicating that different detectors tend to identify similar challenging regions of the

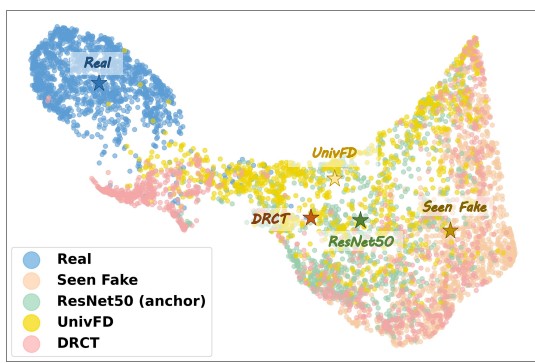

*Figure 6.* **Visualization of boundary-induced fake samples guided by different detectors.** We project boundary-induced samples generated using different detectors separately (ResNet50, UnivFD, and DRCT) into the feature space of the ResNet50 detector. Despite being guided by different critics, the resulting samples largely overlap, indicating that these detectors uncover similar challenging regions of the generative space.

*Table 9.* **Impact of generator choice in PROBE.** We replace SD 1.4 with SD 2.1 and SD 3.5-Medium for boundary exploration, and report the average bAcc of the refined detector across seven benchmarks. The best result is in **bold**. PROBE remains effective across all three generators and consistently outperforms DRCT.

| Method | ResNet50 | PROBE (SD 1.4) | PROBE (SD 2.1) | PROBE (SD 3.5-M) | DRCT |
|---|---|---|---|---|---|
| **Avg bAcc. (%)** | 63.5 | **78.1** | 77.8 | 76.2 | 70.9 |

generative space. To further study this, Table 7 compares detector performance when trained with boundary-induced samples from a single detector versus samples aggregated from multiple detectors. While incorporating samples from multiple detectors yields slight improvements, the gains are marginal compared to using samples generated with a single detector. This is consistent with the visualization in Figure 6 and suggests that a single detector can uncover the dominant hard regions of the generative space.

**Impact of probing generator.** We investigate the role of the generator used in PROBE for boundary exploration. We first study PROBE's sensitivity to the choice of probing generator. Following the PROBE-ResNet50 setup in Section 4.1, we replace SD 1.4 with SD 2.1 and SD 3.5-Medium (Esser et al., 2024), the latter being a more recent generator built on a different architecture (DiT) (Peebles & Xie, 2022). As shown in Table 9, PROBE remains stable across all three generators and consistently outperforms DRCT. This indicates that PROBE is not tied to a specific generator architecture, and that its effectiveness stems from exploring challenging generative variations rather than from properties of any particular probing model.

We further study whether aggregating boundary-induced samples from multiple generators yields gains. Using the

*Table 10.* **Impact of aggregating boundary-induced sample from multiple generators.** We aggregate boundary-induced sample from multiple generators, and report the average bAcc of the refined detector across seven benchmarks. Wo consider three sets of PROBE samples with identical data volume: (i) 1G: PROBE samples from SD 1.4, (ii) 2G: SD 1.4 and SD 2.1, and (iii) 3G: SD 1.4, SD 2.1 and SD 3.5-Medium. The best result is in **bold**. Combining boundary-induced samples from multiple generators consistently improves detector performance.

| Method | ResNet50 | PROBE (1G) | PROBE (2G) | PROBE (3G) | DRCT |
|---|---|---|---|---|---|
| **Avg bAcc. (%)** | 63.5 | 78.1 | 78.9 | **80.2** | 70.9 |

ResNet50 detector, we combine boundary-induced samples from SD 1.4, SD 2.1, and SD 3.5-Medium. In Table 10, aggregation brings consistent improvements over single-generator probing, suggesting that multi-generator probing is a viable path toward stronger detector generalization.

## 6. Conclusion

In this work, we study the generalization challenge in AI-generated image detection and identify a key limitation: training data often covers only a limited portion of the generative space. As a result, detectors tend to overfit to seen generators and struggle with unseen ones. To address this issue, we propose PROBE, which uses the detector as a critic to guide the generator toward challenging yet realistic samples. These boundary-induced samples expose failure cases that are difficult to obtain through standard sampling and help improve detector robustness. Extensive experiments demonstrate that PROBE consistently enhances generalization across multiple benchmarks. We hope this work offers a practical perspective on improving detector generalization.

**Limitations and Future Work.** Boundary exploration is currently performed using a single diffusion model, which provides a practical way to expose challenging samples. Incorporating generators from different model families could further improve coverage of diverse generation patterns. In addition, the current framework assumes access to a modifiable generator during detector refinement. Extending PROBE to fully black-box settings remains an interesting direction for future work.

## Acknowledgment

This work is supported in part by the National Natural Science Foundation of China (NSFC) under Grant No.62376292, Guangdong Provincial General Fund No. 2024A1515010208, and Guangzhou Science and Technology Program Project No.2025A04J5465, 2024A04J6365. We gratefully acknowledge this support. We also thank the anonymous reviewers for their insightful comments that helped improve this paper.

## Impact Statement

This paper presents work whose goal is to advance the field of Machine Learning. There are many potential societal consequences of our work, none which we feel must be specifically highlighted here.

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

The Appendix provides additional details of our work. Section A provides the detailed derivation of the efficient optimization procedure based on Deep Reward Tuning. Section B describes the implementation details of the detector and generator. Section C reports the computational cost. Section D analyzes the generalization mechanism of PROBE. Section E summarizes the evaluation datasets. Section F presents additional quantitative results of main experiments. Section G provides analysis on hyperparameter. Finally, Section H shows qualitative visualizations of PROBE samples and realistic-generation images.

## A. Efficient Optimization via Deep Reward Tuning

In this section, we introduce the backpropagation of the total loss $\mathcal{L}$ defined in Equation (1). Since directly backpropagating through the full diffusion trajectory demands high GPU memory (Prabhudesai et al., 2023), we adopt Deep Reward Tuning (Wu et al., 2024) to fine-tune generator more efficiently.

Denoising in the diffusion model is an iterative process, the result $\mathbf{x}_{t-1}$ at the next time step $t-1$ is obtained by denoising the result $\mathbf{x}_t$ at the current step $t$:

$$\mathbf{x}_{t-1} = a_t \mathbf{x}_t + b_t \epsilon_\phi(\mathbf{x}_t, t) + c_t \epsilon. \tag{3}$$

where $\epsilon \sim \mathcal{N}(0, I)$ is random Gaussian noise, and $a$, $b$, $c$ are coefficients determined by the sampling algorithm. Therefore, the gradient of $\mathcal{L}$ is calculated as follows:

$$\nabla_\phi \mathcal{L} = \sum_{t=0}^{T} \frac{\partial \mathcal{L}}{\partial \mathbf{x}_t} \cdot \frac{\partial \mathbf{x}_t}{\partial \phi}. \tag{4}$$

where $\phi$ is the parameter to be optimized. However, backpropagating gradients through all timesteps demands extremely high GPU memory (Prabhudesai et al., 2023). For this issue, we block the gradients to the input of the denoising network and training only specific time steps. At this point, Equation (3) is modified to:

$$\mathbf{x}_{t-1} = a_t \mathbf{x}_t + b_t \epsilon_\phi(\mathbf{sg}(\mathbf{x}_t), t) + c_t \epsilon. \tag{5}$$

sg denotes the stop gradient operation. After blocking the gradient on the input $x_t$, the gradients between neighboring time steps satisfy:

$$\partial \mathbf{x}_{t-1} = a_t \partial \mathbf{x}_t.$$

Thus, the gradient of $x_t$ can be calculated as

$$\partial \mathbf{x}_t = \prod_{s=1}^{t} a_s^{-1} \partial \mathbf{x}_0.$$

This simplifies the gradient computation to efficient scalar multiplications and allows us to sample a subset of sampling steps for training. Accordingly, we uniformly sample $K$ training steps $T_{train} := \{t_s, t_s + \lfloor \frac{T}{K} \rfloor, \ldots, t_s + K \lfloor \frac{T}{K} \rfloor\}$, $t_s$ is a random start timestep that ensures $T_{train} \subseteq 1, \ldots, T$. Training only sampling steps in $T_{train}$ accelerates convergence while reducing computational overhead. Based on this, $\mathcal{L}$ can be calculated using the following formula:

$$\nabla_\phi \mathcal{L} = \sum_{t \in T_{train}} \left( \frac{\partial \mathcal{L}}{\partial \mathbf{x}_0} \cdot \prod_{s=1}^{t} a_s \right) \cdot \frac{\partial \mathbf{x}_t}{\partial \phi}. \tag{6}$$

which allows gradients to guide the generation process without backpropagating through every diffusion step. We apply LoRA (Hu et al., 2022) to the generator's U-Net and employ gradient checkpointing to further reduce memory consumption.

## B. Implementation Details

### B.1. Details of Detector Implementation

**Model Architecture.** We employ two distinct architectures as backbones to construct our baseline detectors: the CNN-based ResNet-50 (He et al., 2016) and the Transformer-based DINOv2-ViT-L (Oquab et al., 2024). A linear classifier is attached to each backbone to perform binary classification. The input resolutions are set to 224×224 for ResNet50 and 336×336 for DINOV2. For both training and inference, we extract patches of the corresponding resolution via cropping rather than resizing; padding is applied to images smaller than the target size. At inference time, for large images, we average the logits over multiple patches to obtain the final prediction, following (Guillaro et al., 2025).

**Data Augmentation.** To enhance the robustness of the detectors, we incorporate a suite of standard data augmentations during both the pre-training and fine-tuning stages. The augmentation pipeline includes:

- Random JPEG Compression: Quality factor sampled from $[50, 100]$.
- Random Gaussian Blur: Sigma sampled from $[0, 3]$.
- Random Gaussian Noise: Standard deviation sampled from $[0, 55]$.
- Random Resizing: Scale factor sampled from $[0.5, 2.0]$.
- Color Jittering.
- Geometric Transformations: Random cropping, flipping, and rotation.

**Training Baseline Detectors.** We train the baseline detectors using standard optimization protocols:

- ResNet-50: Trained on the GenImage SD1.4 split (Zhu et al., 2023b). We utilize the AdamW optimizer with a weight decay of 1e-4, a learning rate of 1e-4, and a batch size of 128.
- DINOv2: Trained on the Reconstruction Training Set (Guillaro et al., 2025). We utilize the AdamW optimizer with a weight decay of 1e-5, a learning rate of 1e-5, and a batch size of 32.

Both models are optimized using the Binary Cross-Entropy (BCE) loss function.

**Fine-tuning Detectors.** To improve the generalization capability of our detectors, we further fine-tune the baselines using PROBE samples. We construct a fine-tuning dataset, $D_{\text{PROBE}}$, by combining PROBE samples with an equal number of real images from COCO2014 (Lin et al., 2014) that share corresponding captions. The detectors are fine-tuned on a mixture of the original pre-training dataset $D_{\text{pre}}$ and $D_{\text{PROBE}}$. Specifically, within each iteration, we sample mini-batches of equal size from both $D_{\text{pre}}$ and $D_{\text{PROBE}}$. The final loss is computed as a weighted sum:

$$\mathcal{L}_{\text{total}} = (1 - w)\mathcal{L}_{\text{pre}} + w\mathcal{L}_{\text{PROBE}}$$

where $\mathcal{L}_{\text{pre}}$ and $\mathcal{L}_{\text{PROBE}}$ denote the losses computed on the respective batches. We set the weighting hyperparameter $w = 0.5$ for all experiments. For ResNet 50, we utilize the AdamW optimizer with a weight decay of 1e-5, a learning rate of 1e-5, and a batch size of 64. For DINOv2, we utilize the AdamW optimizer with a weight decay of 1e-6, a learning rate of 1e-6, and a batch size of 16. We utilize Binary Cross-Entropy loss for optimization. Following (Chen et al.), we evaluate the balanced accuracy on all datasets at the end of each epoch and employ early stopping to prevent overfitting.

## B.2. Details of Generator Fine-tuning

**Model Configuration.** We select Stable Diffusion 1.4 and Stable Diffusion 2.1 (Rombach et al., 2022) as our target generators for fine-tuning, which align with the seen generators of baseline detectors. For image synthesis, we employ the DDIM sampler (Song et al.) with 35 sampling steps. The classifier-free guidance scale is set to 7.5, and the output resolution is fixed at 512×512. To achieve efficient and controllable fine-tuning, we incorporate lightweight trainable parameters using Low-Rank Adaptation (LoRA) (Hu et al., 2022). LoRA modules with a rank of 128 are applied to the attention layers of the diffusion U-Net (Qin et al., 2020).

**Optimization and Training Strategy.** The fine-tuning process is guided by the first 20,000 prompts from the COCO2014 (Lin et al., 2014). We set the perceptual loss weight to $\lambda = 1.0$ to balance the visual realism of the generated samples against their detection difficulty. We fine-tune the models for a single epoch using the AdamW optimizer with a weight decay of 1e-6, a learning rate of 1e-5, and a batch size of 16. Besides, we adopt the Deep Reward Tuning (Wu et al., 2024) to optimize the generation process efficiently. To balance the trade-off between fine-tuning performance and computational overhead, we configure the hyperparameters as follows: $T = 35$, $K = 5$, and $t_s = 5$. Furthermore, gradient checkpointing is enabled to further reduce GPU memory consumption.

Throughout the training process, we store the generated images corresponding to the 20k prompts. These images serve as boundary-induced samples and are subsequently utilized to construct the dataset $D_{\text{PROBE}}$ for fine-tuning the detectors.

All experiments are conducted on a server equipped with two NVIDIA A800 (40GB) GPUs.

*Table 11.* **Computational cost for each stage of PROBE (in GPU hours, measured on NVIDIA A800 40GB).**

| Stage | ResNet50 (SD 1.4) | DINOv2 (SD 2.1) |
|---|---|---|
| Baseline pretraining | 3 h | 10 h |
| PROBE: boundary exploration + sample generation | 25 h | 30 h |
| PROBE: detector fine-tuning | 6 h | 28 h |

## C. Computational Overhead

Table 11 summarizes the training cost of each stage on NVIDIA A800 (40GB). PROBE introduces additional cost beyond baseline pretraining due to boundary exploration; however, this overhead is moderate in practice.

First, the cost is dominated by image generation rather than optimization. For ResNet50, generating 20k boundary-induced samples with PROBE takes approximately 25 GPU-hours, while standard SD 1.4 sampling of the same number already requires around 20 GPU-hours, the guided optimization adds only ∼5 GPU-hours, thanks to Deep Reward Tuning and single-round probing. Second, PROBE is significantly more data-efficient than comparable methods. DRCT (Chen et al., 2024) and BFree (Guillaro et al., 2025) require 236k and 258k generated samples, respectively, whereas PROBE achieves stronger generalization with only 20k samples. Finally, the cost of PROBE is a one-time offline process. At inference time, the detector architecture remains unchanged and introduces no additional latency.

## D. Generalization Mechanism of PROBE

PROBE performs boundary exploration using a single seen generator, yet consistently improves detection performance on other unseen generators. This does not require covering the full distribution of unseen generators. Instead, PROBE exposes regions of uncertainty under realism constraints, encouraging the detector to rely less on generator-specific cues and more on transferable features. We hypothesize that different generators produce images with common perceptual ambiguities (*e.g.* texture inconsistency, oversmoothing, and semantic mismatch) and that steering generation toward these uncertain regions while maintaining realism uncovers hard regions shared across generators. We provide two supporting analyses and examine a failure case that reveals the limits of single-generator probing.

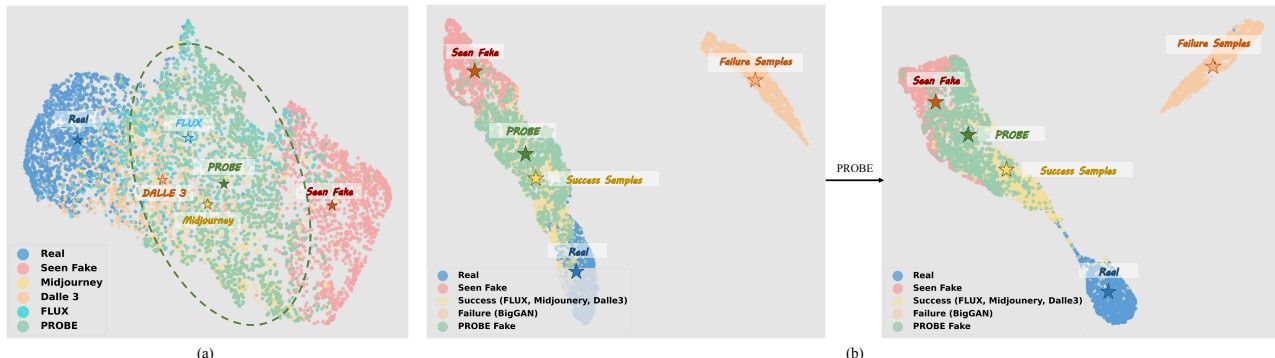

*Figure 7.* **(a) Feature space visualization of the baseline ResNet50 detector.** PROBE boundary-induced samples exhibit substantial overlap with those from architecturally diverse unseen generators (DALL·E 3, Midjourney, and FLUX), which indicates that PROBE effectively captures shared detector-relevant hard regions. **(b) Failure case visualization.** BigGAN samples exhibit significant deviation from PROBE samples, resulting in persistent difficulties for the fine-tuned detector to accurately distinguish these samples.

**Feature-space overlap with unseen generators.** We project architecturally different unseen generators (DALL·E 3 and Midjourney samples from Synthbuster, FLUX samples from AIGI-Quality-Paradox), and PROBE samples into the feature space of the baseline ResNet50 detector via t-SNE (van der Maaten & Hinton, 2008). As shown in Figure 7(a), PROBE samples exhibit substantial overlap with those from unseen generators, indicating that the hard regions identified by PROBE reflect shared ambiguities rather than generator-specific patterns. Refining the detector on these samples reshapes its decision boundary and improves generalization.

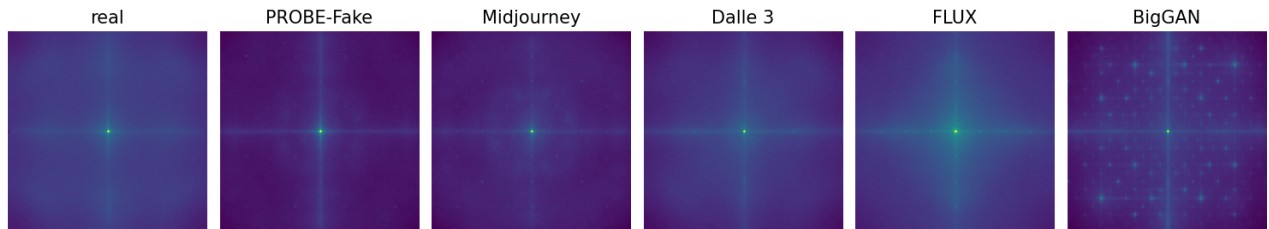

*Figure 8.* **Energy spectra of denoising residuals of images from different sources.** Midjourney, FLUX, DALL·E 3, and PROBE samples exhibit similar spectral patterns, while BigGAN samples show a clearly different grid-like structure.

**Spectral analysis.** Following (Corvi et al., 2023), we extract noise patterns from generated images using a denoising network and analyze their energy spectra. As shown in Figure 8, PROBE samples and those from unseen generators exhibit similar spectral patterns, providing additional evidence that PROBE captures shared generative characteristics rather than artifacts specific to the probing generator.

**Failure case analysis.** We identify BigGAN as a case where the PROBE-ResNet50 detector shows limited improvement. As shown in Figure 7(b), BigGAN forms an isolated cluster in the detector's feature space with minimal overlap with PROBE samples, and its frequency spectrum exhibits a distinct grid-like structure absent in other unseen generators. This suggests that gains are limited when an unseen generator lies outside the explored regions. Notably, PROBE-DINOv2 achieves 95% accuracy on the BigGAN subset of GenImage, suggesting that stronger backbones with richer pretrained representations can help alleviate this limitation.

# E. Details of Datasets

*Table 12.* **Datasets overview, including the number of real and fake images, the sources of real images, and the number of generators.**

| Datasets | # Fake/Real | Real Sources | # Generators |
|---|---|---|---|
| GenImage (Zhu et al., 2023b) | 48k / 48k | ImageNet | 8 |
| Synthbuster (Bammey, 2023) | 9k / 1k | RAISE | 9 |
| AIGI-Quality-Paradox (Xiao et al., 2025) | 24k / 4k | SA-1B, LAION, COCO, CC3M | 6 |
| Chameleon (Yan et al., 2025a) | 11.2k / 14.9k | Internet | Unknown |
| SynthWildX (Cozzolino et al., 2024) | 1.5k / 0.5k | X | 3 |
| WildRF (Cavia et al., 2024) | 1.25k / 1.25k | Reddit, FB, X | Unknown |
| AIGI-Bench (Li et al., 2026) | 9k / 9k | Reddit, FB, X | Unknown |

We provide statistics of the datasets utilized for evaluation in Table 12. These datasets encompass a wide array of mainstream generators and exhibit high diversity in terms of both content and format, thereby minimizing evaluation bias.

# F. Additional Results

Following (Wang et al., 2020; Ojha et al., 2023; Tan et al., 2024; Yan et al., 2025b; Guillaro et al., 2025), we additionally report Average Precision (AP) scores in Table 13. AP serves as a threshold-independent metric to evaluate the detector's discriminative capability across all possible decision boundaries. The results yield two key observations: 1) PROBE delivers consistent improvements over baselines. Specifically, the mean AP increases by 11.2% for ResNet-50 and 4.0% for DINOv2. This suggests that PROBE not only optimizes the decision boundary but also enhances the discriminative capability of the learned features, rendering the detector more robust to varying decision thresholds; 2) PROBE-DINOv2 achieves state-of-the-art performance. It attains an impressive mean AP of 97.8%, ranking highest across all evaluated datasets. Notably, even on top of strong baselines that already exhibit high precision, PROBE achieves further gains, highlighting its efficacy in distinguishing hard samples.

*Table 13.* **Overall comparison across seven benchmarks.** The evaluation metric is Average Precision (AP%). The methods are grouped into two categories based on their training data. For each group, the best result is marked in **bold** and the second best is underlined.

| Method | GenImage | AIGI-Quality-Paradox | Synthbuster | Chameleon | SynthWildX | WildRF | AIGI-Bench | Avg. |
|---|---|---|---|---|---|---|---|---|
| *Trained on GenImage SD 1.4* | | | | | | | | |
| CNNSpot (Wang et al., 2020) | 82.6 | 74.5 | 83.6 | 57.7 | 62.2 | 73.0 | 58.5 | 70.3 |
| UnivFD (Ojha et al., 2023) | 88.6 | 78.7 | **90.4** | 56.0 | 61.5 | 61.7 | 60.5 | 71.1 |
| NPR (Tan et al., 2024) | 84.1 | 64.7 | 48.4 | 48.3 | 55.5 | 67.6 | 64.0 | 61.8 |
| DRCT (Chen et al., 2024) | **97.3** | 81.8 | 74.2 | 67.8 | **84.8** | 86.2 | 79.2 | 81.6 |
| AIDE (Yan et al., 2025a) | 85.1 | 71.3 | 64.4 | 64.2 | 66.0 | 77.0 | 72.6 | 71.5 |
| Effort (Yan et al., 2025b) | 93.7 | 81.4 | 87.7 | 54.9 | 61.3 | 85.5 | 66.0 | 75.8 |
| ResNet50 | 87.6 | 81.0 | 67.7 | 56.5 | 73.2 | 82.2 | 66.5 | 73.5 |
| **PROBE (ResNet50)** | 89.7 (+2.1%) | **94.8 (+13.8%)** | 81.5 (+13.8%) | **75.1 (+18.6%)** | 82.9 (+9.7%) | **89.2 (+7.0%)** | 79.8 (+13.3%) | **84.7 (+11.2%)** |
| *Trained on Reconstruction Training Set* | | | | | | | | |
| BFree (Guillaro et al., 2025) | 97.3 | 96.3 | 99.0 | 82.1 | 98.5 | 97.9 | 92.4 | 94.8 |
| DDA (Chen et al.) | 98.4 | 96.7 | 96.2 | 87.6 | 95.4 | 94.9 | 91.5 | 94.4 |
| DINOv2 | 95.9 | 91.7 | 97.3 | 85.6 | 98.2 | 97.1 | 90.5 | 93.8 |
| **PROBE (DINOv2)** | **99.7 (+3.8%)** | **97.7 (+6.0%)** | **99.6 (+2.3%)** | **92.9 (+7.3%)** | **98.7 (+0.5%)** | **99.6 (+2.5%)** | **96.5 (+6.0%)** | **97.8 (+4.0%)** |

*Table 14.* **Sensitivity to the perceptual regularization weight** $\lambda$. We report average balanced accuracy (%) across seven benchmarks and HPSv3 score. The setting used in our main experiments is highlighted.

| $\lambda$ | baseline | 0.0 | 0.25 | 0.5 | 0.75 | **1.0** | 1.5 | 2.0 |
|---|---|---|---|---|---|---|---|---|
| **Avg bAcc.** | 63.5 | 74.7 | 77.0 | 77.1 | 76.9 | **78.1** | 75.9 | 76.1 |
| **HPSv3 Score** | / | 0.01 | 2.52 | 3.38 | 3.94 | 4.37 | 4.67 | **4.88** |

# G. Analysis on Hyperparameter

We provide sensitivity analysis on four hyperparameters of PROBE: (1) the weight $\lambda$ of the perceptual regularization, (2) the rank of the LoRA modules applied to the diffusion U-Net, (3) the number of PROBE samples used to fine-tune the detector, and (4) the number of boundary exploration and detector adaptation process iterations. All experiments are conducted with the ResNet50 detector. We report average balanced accuracy across the seven benchmarks introduced in Section 4.1, together with the HPSv3 score (Ma et al., 2025) of PROBE samples to quantify visual quality and text-image alignment.

**Perceptual regularization weight** $\lambda$. $\lambda$ controls the trade-off between boundary exploration and visual realism in Equation (1). As shown in Table 14, removing perceptual regularization ($\lambda = 0$) yields substantially lower detector performance compared to other settings, indicating that constraining PROBE samples to remain realistic benefits not only visual fidelity but also detection accuracy. Among the remaining settings, increasing $\lambda$ monotonically improves the HPSv3 score, while detector performance peaks at $\lambda = 1.0$ and degrades slightly at larger values, reflecting a balance between realism and the boundary exploration. We therefore use $\lambda = 1.0$ in main experiments.

**LoRA rank.** To investigate the sensitivity of PROBE to LoRA rank, we vary the rank of the LoRA modules from 16 to 128. Table 15 shows that detector performance varies within roughly two points across this range, and HPSv3 scores remain in a similarly narrow band. This indicates that PROBE is robust to the choice of LoRA rank. We adopt rank 128 in our main experiments, where it gives the best average bAcc.

**Number of PROBE samples.** As shown in Table 16, detector performance improves consistently as the sample count grows from 5k to 40k, while the marginal gain decreases at larger scales (e.g., +12.2% from $0 \rightarrow 5k$ versus +0.8% from $20k \rightarrow 40k$). We therefore use 20k PROBE samples in our main experiments as a practical balance between performance and computational cost.

**Iterative probing.** We investigate whether repeating the boundary exploration and detector adaptation process across multiple iterations yields further gains. At each iteration, the generator is re-optimized using the detector refined in the previous round. As shown in Table 17, the average bAcc improves from 63.5% (baseline) to 78.1% after the first iteration, and further to 79.4% and 80.1% after the second and third iterations, with marginal improvements of +1.3% and +0.7%. These results suggest that iterative probing can bring additional gains, but the first iteration already captures the dominant improvement. We therefore adopt a single-round strategy in our main experiments as a practical trade-off between performance and computational cost.

*Table 15.* **Sensitivity to the LoRA rank.** We report average balanced accuracy (%) across seven benchmarks and HPSv3 score. The setting used in our main experiments is highlighted.

| LoRA rank | baseline | 16 | 32 | 64 | **128** |
|---|---|---|---|---|---|
| **Avg bAcc.** | 63.5 | 76.3 | 77.3 | 77.0 | **78.1** |
| **HPSv3 Score** | / | 3.01 | 4.24 | 4.15 | **4.37** |

*Table 16.* **Sensitivity to the number of PROBE samples.** We report average balanced accuracy (%) across seven benchmarks. The setting used in our main experiments is highlighted.

| Sample number | 0 | 5k | 10k | **20k** | 40k |
|---|---|---|---|---|---|
| **Avg bAcc.** | 63.5 | 75.7 | 77.3 | 78.1 | **78.9** |

## H. Qualitative Cases

### H.1. Visualization of Boundary-induced Fake Samples

We visualize the Boundary-induced Fake Samples synthesized using various detectors as critics in Figure 9. In contrast to the PGD (Madry et al., 2017) adversarial examples shown in (b), which deviate from the natural image manifold, the variants explored by PROBE within the generative space preserve visual realism while effectively evading detection.

### H.2. Visualization of SD 1.5-Realistic-Vision Images

We present the images generated by SD 1.5 (Rombach et al., 2022) and SD 1.5-Realistic-Vision (SG_161222, 2024) in Figure 10. Images from SD 1.5-Realistic-Vision exhibit significant differences from those of SD 1.5 in terms of lighting, texture, and content semantics; these highly realistic images not only pose greater challenges to detectors but also highlight the importance of model robustness to the realistic generation shift.

*Table 17.* **Performance across probing iterations.** We report average balanced accuracy (%) across seven benchmarks. The setting used in our main experiments is highlighted.

| Number of iterations | baseline | **iteration 1** | iteration 2 | iteration 3 |
|---|---|---|---|---|
| **Avg bAcc.** | 63.5 | 78.1 | 79.4 | **80.1** |
| **Improvement over the previous iteration** | / | **+14.6** | +1.3 | +0.7 |

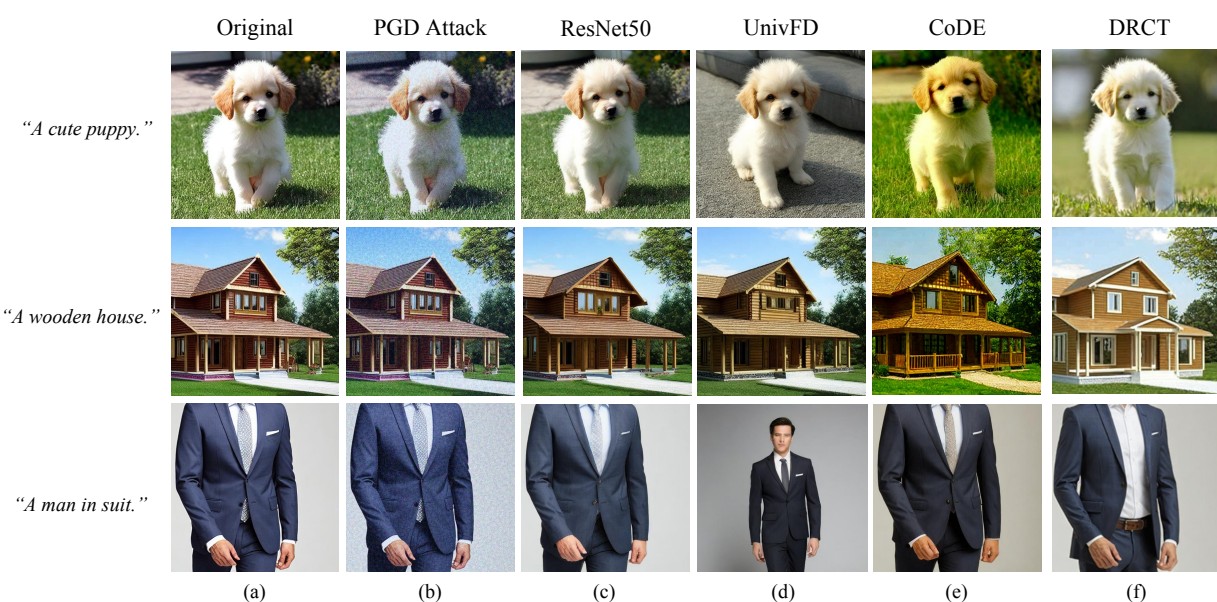

|  Original | PGD Attack | ResNet50 | UnivFD | CoDE | DRCT |
|---|---|---|---|---|---|
| (a) | (b) | (c) | (d) | (e) | (f) |

*Figure 9.* **Qualitative results of boundary-induced fake samples using various detectors as critics.** Each row shows images corresponding to: the original SD 1.4 images, PGD adversarial attack examples, and boundary-induced fake samples from the generator steered by ResNet50, UnivFD, CoDE, and DRCT, respectively. PROBE samples remain realistic while effectively evading detection.

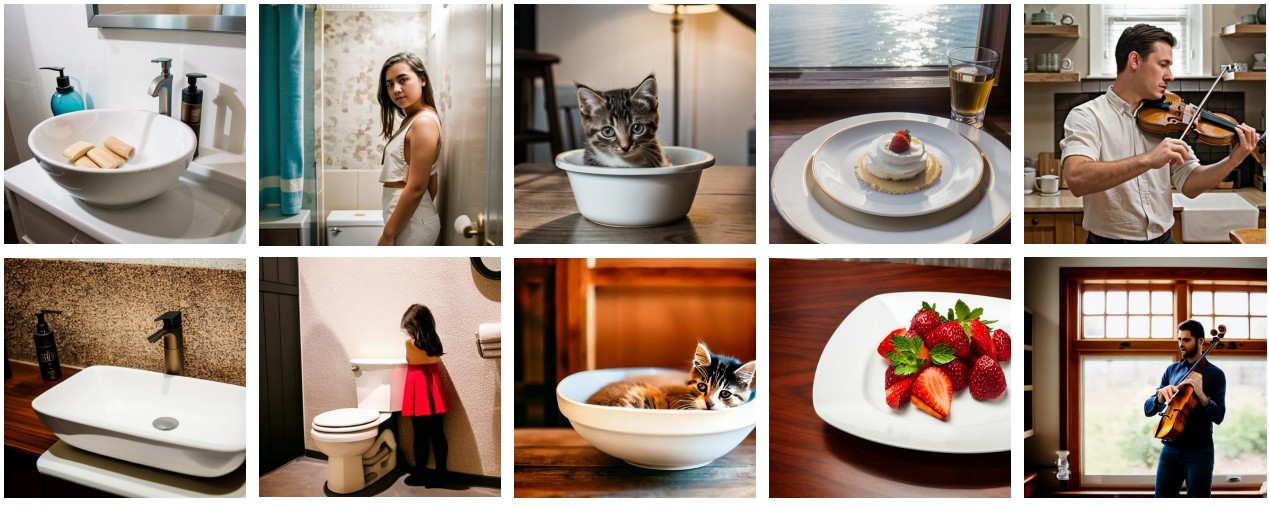

*Figure 10.* **Visualization of SD 1.5 and SD 1.5-Realistic-Vision images.** Images from SD 1.5-Realistic-Vision (first row) are more visually realistic than SD 1.5(second row) and thus pose greater challenges to detectors.

