# OpenReview forum: "Where Detectors Fail: Probing Generative Space for Generalizable AI-Generated Image Detection"
_ICML.cc/2026/Conference — ICML 2026 regular_

### Official Review · Reviewer_HUVf · 2026-03-10

**Soundness:** 3
**Presentation:** 3
**Significance:** 3
**Originality:** 3
**Overall Recommendation:** 5
**Confidence:** 3

**Summary:**

This paper examines poor cross-generator generalization of AI-generated image detectors and attributes it partly to limited coverage of generator latent or manifold variations in training. The proposed method PROBE uses the detector as a critic to LoRA-fine-tune a diffusion generator, then retrains the detector on them.

**Compliance With Llm Reviewing Policy:**

Affirmed.

**Final Justification:**

The authors have addressed most of my concerns and the significance of the work ought to be recognized.

**Key Questions For Authors:**

1. How sensitive are results to the perceptual regularization weight λ and the LoRA rank? Could you include quantitative ablations to surface the realism–hardness trade-off?
2. Beyond a single seen generator, how does aggregating boundary-induced samples from multiple seen generators affect generalization and compute?
3. What is the effect of varying the number of probing prompts on the final detector performance and compute cost?

**Limitations:**

yes

**Strengths And Weaknesses:**

Strengths:

1. The work reframes training data construction for AIGI detection as active search in the generator manifold guided by the detector decision boundary, moving beyond standard sampling and pixel-space adversarial examples.

2. Comprehensive evaluation on seven benchmarks (mixing in-house and in-the-wild) with two training regimes (GenImage-trained and reconstruction-trained) and two detector backbones shows consistent gains.

Weaknesses:

1. Although the method empirically improves generalization to unseen generators, it remains unclear how coverage of the generative space transfers across fundamentally different families or model customizations beyond the provided benchmarks.
2. Sensitivity to the perceptual regularization weight $\lambda$ and LoRA rank is only illustrated qualitatively. Quantitative ablations would help assess stability and distinguish realism constraints from boundary-exploration pressure.
3. Closely related work on detector-guided diffusion for generating hard negatives in other domains (e.g., DADA for medical detection) and detector-reward based generator optimization (e.g., RealGen) is not discussed.

---

> ### Author Rebuttal · Authors · 2026-03-31
>
> >**Q1: Explanation of PROBE's Generalization Principle on Unseen Generators.**
>
> PROBE improves the detector's decision boundary by exposing regions of uncertainty under realism constraints, thereby enhancing generalization to unseen generators.
> Our hypothesis is that different generators, despite architectural differences, produce images with common perceptual ambiguities such as texture inconsistency, oversmoothing, and semantic mismatch. By steering generation toward uncertain regions while maintaining realism, PROBE encourages the detector to rely less on generator-specific cues and more on transferable features.
>
> To support this, we project samples from architecturally diverse unseen generators—DALL·E 3, Midjourney and FLUX—along with PROBE boundary-induced samples into the feature space of the baseline ResNet50 detector, visualized via t-SNE. The results reveal a significant overlap between PROBE samples and those from unseen generators, suggesting shared detector-relevant hard regions.
>
> ||BigGAN|GauGAN|CycleGAN|ProGAN|StyleGAN|StyleGAN2|NOVA|Avg Acc.
> -|-|-|-|-|-|-|-|-
> DINOv2 baseline|85.1|96.1|65.4|93.0|78.1|74.0|94.2|82.1
> PROBE-DINOv2|**96.0**|**99.0**|**83.4**|**97.3**|**91.0**|**90.4**|**99.8**|**91.4**
>
> We further evaluate PROBE on generators from entirely different families, including GAN subsets from AIGCDetectBenchmark [a] and an autoregressive generator (NOVA) from EvalGEN [b]. PROBE-DINOv2 achieves 91.4% average accuracy (+9.3% over the DINOv2 baseline), suggesting that with a strong backbone and rich pretrained representations, PROBE can generalize to generators from different families.
>
> [a] PatchCraft: Exploring Texture Patch for Efficient AI-generated Image Detection, ArXiv2023.
>
> [b] Dual Data Alignment Makes AI-Generated Image Detector Easier Generalizable, NeurIPS2025.
>
> >**Q2:Sensitivity to the perceptual regularization weight λ and LoRA rank.**
>
> Thank you for this suggestion. We report quantitative ablations on λ and LoRA rank below (average bAcc across 7 benchmarks):
>
> λ|baseline|0.0|0.25|0.5|0.75|1.0|1.5|2.0|
> -|-|-|-|-|-|-|-|-|
> Avg bAcc.|63.5|74.7|77.0|77.1|76.9|**78.1**|75.9|76.1|
> HPSv3 Score|-|0.01|2.52|3.38|3.94|4.37|4.67|**4.88**|
>
> LoRA rank|baseline|16|32|64|128|
> -|-|-|-|-|-|
> Avg bAcc.|63.5|76.3|77.3|77.0|**78.1**|
> HPSv3 Score|-|4.01|4.24|4.15|**4.37**|
>
> Perceptual regularization consistently improves both visual quality of PROBE samples and detector performance. As λ increases, image quality improves monotonically, while detector performance peaks at λ = 1.0, reflecting a balance between realism and boundary exploration.
>
> The impact of LoRA rank is relatively small, indicating PROBE is not sensitive to this parameter across a range of settings.
>
> >**Q3: Discussion on DADA and RealGen.**
>
> Thank you for pointing out these related works. While both share the high-level idea of using detector feedback to guide generation, PROBE differs from them in objective and mechanism.
>
> DADA injects adversarial perturbations during the denoising process to synthesize hard false positives for polyp detection, without modifying the generator's parameters. RealGen uses detector-based rewards to optimize generators toward higher photorealism, with the goal of improving generation quality rather than detection.
>
> In contrast, PROBE steers the generator in the generative space to expose detector failure modes, with the goal of improving cross-generator generalization rather than enhancing realism or targeting a specific detection task.
>
> We will include both works in the related work section and clarify these differences.
>
> >**Q4: Impact of Aggregating Multiple Generators on Detector Performance.**
>
> Following the suggestion, we conducted experiments by aggregating boundary-induced samples from multiple generators using the ResNet50 detector:
>
> |PROBE Samples Source|Cost (GPU-hours)|Avg bAcc.|
> |-|-|-|
> |w/o PROBE samples|-|63.5|
> |SD1.4|25|78.1|
> |SD1.4+SD2.1|25+29=54|78.9|
> |SD1.4+SD2.1+SD3.5M|25+29+30=84|**80.2**|
>
> Aggregating PROBE samples from additional generators yields further improvements (+0.8, +1.3), while the probing cost scales from 25 to 85 GPU-hours.  These results suggest that multi-generator probing offers a viable path for further improvement when additional compute is available.
>
> >**Q5: Impact of the Number of Probing Prompts on Detector Performance and Computational Cost.**
>
> We observe that increasing the number of probing prompts consistently improves detector performance, while even a relatively small number already provides substantial gains.
>
> Prompt number|0|5k|10k|20k|40k|
> -|-|-|-|-|-|
> Avg bAcc.|63.5|75.7|77.3|78.1|**78.9**|
>
> Since the generator is updated for one epoch, computational cost scales approximately linearly with prompt count. However, the marginal gain diminishes at larger scales (e.g., +12.2 from 0→5k vs. +0.8 from 20k→40k). We therefore choose 20k prompts in the main experiments as a practical balance between performance and cost.

---

> > ### Author Rebuttal · Reviewer_HUVf · 2026-04-04
> >
> > Thank you for the response. After carefully reviewing all feedback, I have decided to raise my original score.

---

> > > ### Author Response · Authors · 2026-04-04
> > >
> > > Dear Reviewer HUVf,
> > >
> > > Thank you for your positive comments and for raising your score. We are glad our response has addressed your concerns. We acknowledge your constructive suggestions help us improve our work, and we will incorporate the rebuttal in the revision.
> > >
> > > Best,
> > >
> > > The Authors

---

### Official Review · Reviewer_YTGy · 2026-03-11

**Soundness:** 2
**Presentation:** 3
**Significance:** 2
**Originality:** 2
**Overall Recommendation:** 3
**Confidence:** 4

**Summary:**

This paper tackles the generalization failure of AI-generated image detectors to unseen generators. It argues that standard sampling covers only a narrow portion of the generative space and proposes PROBE, a two-stage framework that uses a pre-trained detector as critic to steer a seen diffusion model toward its decision boundary. Perceptual regularization ensures realism. The resulting boundary-induced samples are used to fine-tune the detector in a single round. Experiments on seven benchmarks show consistent gains.

**Compliance With Llm Reviewing Policy:**

Affirmed.

**Final Justification:**

I find the work to be solid overall; however, the level of novelty appears to be limited. Therefore, I will maintain my original score.

**Key Questions For Authors:**

Please see weaknesses.

**Limitations:**

yes

**Strengths And Weaknesses:**

Strengths:

- The paper delivers a strong empirical contribution through comprehensive evaluation across seven diverse benchmarks. It achieves consistent and substantial performance gains across different detector backbones. In addition, thorough ablation studies on backbone sensitivity, robustness to post-processing, and multi-critic sample overlap convincingly demonstrate the method’s robustness and generality.

- The manuscript is exceptionally well presented, featuring excellent figures, clear and precise mathematical formulations, and an honest discussion of the method’s limitations regarding single-generator probing and single-round exploration, all of which significantly enhance readability and facilitate understanding of the proposed framework.

Weaknesses:

- The core idea of using a detector to guide the generation of adversarial or hard examples is not novel. In the specific context of AI-generated image detection, this adversarial paradigm has already been thoroughly explored in GenDet (GenDet: Towards Good Generalizations for AI-Generated Image Detection), which employs teacher-student discrepancy maximization in feature space to synthesize difficult fake samples. Although PROBE applies the idea in generative space through LoRA editing and perceptual regularization, the high-level minimax motivation and the goal of exposing decision-boundary weaknesses remain conceptually similar. As a result, the paper reads more as an engineering refinement of existing adversarial techniques than as the introduction of a fundamentally new algorithm.

- Although PROBE performs all boundary exploration on only a single known generator, the paper’s core argument centers on achieving better coverage of the generative space. However, generating adversarial samples exclusively from one known generator cannot reliably guarantee that the refined decision boundary will generalize to truly unseen generators, since the discovered hard regions may still be generator-specific.

- Generating the boundary-induced adversarial samples introduces substantial additional time overhead.

---

> ### Author Rebuttal · Authors · 2026-03-31
>
> >**Q1: Lack of novelty. Using a detector to guide the generation of adversarial or hard examples is not novel … has already been thoroughly explored in GenDet ... the paper reads more as an engineering refinement of existing adversarial techniques than as the introduction of a fundamentally new algorithm.**
>
> We acknowledge that PROBE shares a high-level goal with prior approaches (e.g., GenDet, Line 134 “Improving training data of detector”) in exposing decision-boundary weaknesses. However, we argue that PROBE is not an engineering refinement, but a different formulation of how such weaknesses are explored and what constitutes a valid training signal.
>
> GenDet synthesizes hard samples in feature space by maximizing teacher–student discrepancy, without relying on an explicit generative model enforcing that samples correspond to realistic images. As a result, the optimization is unconstrained with respect to data realism, and generated samples may not reflect valid instances from the distribution.
>
> In contrast, PROBE explores the generator space of a pretrained diffusion model to produce realistic samples near the detector’s decision boundary. Rather than maximizing discrepancy, it focuses on natural boundary cases, revealing semantic and structural ambiguities that are useful for cross-generator generalization.
>
> Furthermore, GenDet treats sample construction as a byproduct of discrepancy maximization. PROBE instead introduces a closed-loop interaction where the detector guides generation to probe its blind spots, shifting the generator from passive sample construction to active boundary exploration in realistic data space.
>
> We will clarify these differences with prior methods in Line 160.
>
> >**Q2: True generalization to unseen generators. Generating adversarial samples exclusively from one known generator cannot  guarantee that the refined decision boundary will generalize to truly unseen generators, since the discovered hard regions may still be generator-specific.**
>
> PROBE does not aim to cover the full distribution of unseen generators. Instead, it improves the detector’s decision boundary by exposing regions of uncertainty under realism constraints, which enhances its ability to generalize to unseen generators. While generator-specific patterns may exist, the boundary regions identified by PROBE are largely general and reflect shared ambiguities across generators.
>
> To support this, we project samples from architecturally different unseen generators—DALL·E 3 and Midjourney from Synthbuster, FLUX from AIGI-Quality-Paradox—along with PROBE boundary-induced samples into the feature space of the baseline ResNet50 detector, visualized via t-SNE. The results show substantial overlap between PROBE samples and those from unseen generators, indicating shared detector-relevant hard regions with the PROBE samples. Refining the detector with PROBE samples reshapes its decision boundary and improves generalization.
>
> Beyond this, we evaluate PROBE on 7 benchmarks covering diverse diffusion, GAN, and autoregressive generators, as well as in-the-wild benchmarks. PROBE-DINOv2 achieves over 90% average accuracy. We further test on GAN subsets from AIGCDetectBenchmark [a] and an autoregressive generator (NOVA) from EvalGEN [b]. PROBE-DINOv2 achieves 91.4% average accuracy on these architecturally different unseen generators, outperforming BFree (88.4%). It provides further empirical evidence that PROBE improves detector generalization well beyond the single generator used for probing.
>
> ||BigGAN|GauGAN|CycleGAN|ProGAN|StyleGAN|StyleGAN2|NOVA|Avg Acc.|
> -|-|-|-|-|-|-|-|-|
> Bfree|91.9|97.0|**86.1**|95.9|82.1|78.0|99.5|88.4|
> PROBE-DINOv2|**96.0**|**99.0**|83.4|**97.3**|**91.0**|**90.4**|**99.8**|**91.4**|
>
> [a] PatchCraft: Exploring Texture Patch for Efficient AI-generated Image Detection, ArXiv2023.
>
> [b] Dual Data Alignment Makes AI-Generated Image Detector Easier Generalizable, NIPS2025.
>
> >**Q3: Computational Cost of PROBE.**
>
> PROBE introduces additional cost due to boundary exploration, but the overhead is moderate in practice.
>
> First, the cost is dominated by image generation rather than optimization. For ResNet50, generating 20k boundary-induced samples with PROBE takes about 25 GPU-hours, while standard SD 1.4 sampling of the same number already requires around 20 GPU-hours — the guided optimization adds only ~5 GPU-hours, thanks to Deep Reward Tuning and single-round probing strategy.
>
> Second, PROBE is significantly more data-efficient than comparable methods. DRCT and BFree require 236k and 258k generated samples, respectively, whereas PROBE achieves stronger generalization with only 20k samples:
>
> ||PROBE|DRCT|Bfree|
> -|-|-|-|
> Generated samples number|20k|236k|258k|
>
> Finally, the cost of PROBE is a one-time offline process. At inference time, the detector architecture remains unchanged and introduces no additional latency.

---

> > ### Author Rebuttal · Reviewer_YTGy · 2026-04-03
> >
> > Thank the authors for the response.
> >
> > Regarding Q2, my concern remains. The t-SNE visualization showing overlap between PROBE samples and unseen generator samples does not constitute a rigorous justification for generalization. The core issue is that generating adversarial samples from a single known generator lacks a principled guarantee that the discovered boundary regions transfer to truly unseen generators. The additional empirical results are appreciated but do not resolve this fundamental limitation.
> >
> > As this concern is not adequately addressed, I will maintain my current score.

---

> > > ### Author Response · Authors · 2026-04-03
> > >
> > > Dear Reviewer YTGy,
> > >
> > > We thank the reviewer for the comment regarding the lack of a principled guarantee for generalization. We address as follows,
> > >
> > > A complete enumeration of all possible generators is inherently infeasible. Following standard practice, we evaluate generalization across a ***diverse and representative set of generators and benchmarks***, demonstrating that PROBE achieves competitive accuracy and strong generalization.
> > >
> > > - ***Strong empirical generalization across benchmarks.***
> > > As shown in Table 1, our method is evaluated on seven benchmarks covering a wide range of unseen generators. The consistent improvements across these datasets demonstrate strong empirical generalization beyond the generator used during boundary exploration.
> > > - ***Coverage of challenging regions in the generative space.***
> > > In the rebuttal, we show that PROBE samples overlap with representative unseen generator samples in feature space (<https://anonymous.4open.science/r/ICML26-rebuttal-1687-07C8/tSNE-visualization.pdf>). It provides intuitive evidence that our boundary-induced samples expand into challenging regions of the generative space, which are underrepresented in standard training data and are critical for generalization.
> > > - ***Validation across generator families.***
> > > We additionally report results on GAN-based subsets from AIGCDetectBenchmark. The consistent gains on these subsets further demonstrate that our method generalizes beyond diffusion models to different generator families.
> > >
> > > ***On the evaluation protocol and justification***: We respectfully note that if such strong empirical evidence is considered insufficient to support generalization, then similar evaluation protocols widely adopted in ***prior works*** (including those compared in Table 1) ***would also face the same limitation***. Our method follows the same standard benchmarking practice, while consistently achieving stronger performance.
> > >
> > > We hope the reviewer will reconsider their assessment based on the above.
> > >
> > > Regards,
> > >
> > > The Authors

---

### Official Review · Reviewer_m4xY · 2026-03-12

**Soundness:** 2
**Presentation:** 2
**Significance:** 3
**Originality:** 2
**Overall Recommendation:** 4
**Confidence:** 3

**Summary:**

To address the generalization failure of AI-generated image detectors on unseen generators, this paper proposes PROBE, a framework that actively explores challenging regions of the generative space. PROBE utilizes the target detector as a critic to steer the generation process toward the detector's decision boundaries. The generator is efficiently optimized to produce boundary-induced fake samples that effectively expose the detector's failure modes. Concurrently, perceptual regularization is applied to ensure these difficult samples maintain high visual realism. The target detector is then fine-tuned using these samples, reshaping its decision boundary to be robust and less reliant on generator-specific artifacts. Experiments demonstrate that PROBE consistently enhances generalization across seven diverse benchmarks, yielding state-of-the-art detection accuracy

**Compliance With Llm Reviewing Policy:**

Affirmed.

**Ethical Review Concerns:**

I do not identify any ethical concerns that require an additional ethics review.

**Final Justification:**

I believe that the empirical results presented in the paper can contribute to the community and slightly outweigh its weaknesses.

**Key Questions For Authors:**

see weakness section

**Limitations:**

yes

**Strengths And Weaknesses:**

**Strengths**
- **Straightforward and Sound Method**
By utilizing the target detector as a critic to guide the generation process, the paper presents a sound methodological shift from passive data collection to active boundary exploration.
- **Improved Generalization Performance**
The proposed framework effectively enhances cross-generator detection accuracy across seven diverse benchmarks, achieving strong empirical results particularly when integrated with the DINOv2 backbone
- **Extensive experiments**
The paper covers a variety of datasets including in-house and in-the-wild datasets, and includes thorough analyses across multiple dimensions: detector backbone comparisons, robustness to image post-processing, and comparisons with data augmentation techniques.



**Weaknesses**
- **Lack of novelty**
When decomposed, PROBE consists entirely of existing components: using the detector as a reward signal to steer the generator is a variant of adversarial training, LoRA-based fine-tuning is standard (Hu et al., 2022), Deep Reward Tuning is directly adopted from Wu et al. (2024), and VGG-based perceptual regularization originates from Johnson et al. (2016). The proposed framework — generate hard negatives via adversarial optimization, retrain the classifier — is a well-established recipe in adversarial robustness. The paper does not sufficiently discuss how PROBE differs from GAN-era adversarial data augmentation methods
- **Insufficient Explanation of the Generalization Mechanism from Single-Generator Exploration**
The main claim of PROBE is that boundary exploration within a single seen generator improves generalization to unseen generators. However, the paper lacks theoretical or empirical analysis explaining why exploring the generative space of SD 1.4/2.1 transfers to outputs from fundamentally different architectures such as DALL·E 3, Midjourney, and FLUX. Figure 6 demonstrates that different detector critics explore similar regions, but this alone is insufficient to explain cross-generator generalization. (1) An analysis of the shared artifact characteristics across different generators would substantially strengthen the paper. (2) Furthermore, a failure case analysis is needed to clarify under what conditions between seen and unseen generators PROBE succeeds or fails to work properly.
- **Weak Justification for Not Performing Iterative Boundary Exploration**
The authors claim that a single round of boundary exploration is sufficient, yet no supporting experimental evidence is presented in the paper. The assertion that "further iterations tend to revisit similar variations" lacks convincingness without empirical verification. Notably, Section 3.2 states that the benefits of iterative exploration are marginal, while the Limitations section leaves its investigation as future work — these two statements are contradictory. If the authors have not properly verified the effect of iterative exploration, the claim in Section 3.2 should be revised accordingly.
- **Minor weakness about presentation**
Although Table 2 presents main results on individual generator performance, there is no reference to or explanation of this table in the main text other than its caption. The authors should add appropriate cross-references and a brief discussion of these findings in the main text.

---

> ### Author Rebuttal · Authors · 2026-03-31
>
> >**Q1: Lack of novelty. PROBE uses existing components and follows a well-established adversarial robustness recipe (generate hard negatives, retrain classifier). Differences from GAN-era adversarial augmentation are insufficiently discussed.**
>
> While the high-level idea of exploring challenging regions is related to prior efforts such as adversarial training (Line 134, “Improving training data of detector”), PROBE differs in where and how this exploration is performed.
>
> Classical adversarial training operates in input space with local perturbations around existing samples. PROBE instead performs generator-space (distribution-level) exploration, steering a diffusion model to produce realistic samples beyond standard sampling while remaining on the natural image manifold, uncovering semantic and structural failure modes unreachable by input-space perturbations.
>
> Prior GAN-based methods generate hard samples but typically optimize for either difficulty or realism in isolation, so samples are not explicitly constrained to be both realistic and near the detector's decision boundary. PROBE enforces realism-constrained boundary exploration, focusing on realistic ambiguities rather than artificially hard samples.
>
> Therefore, although PROBE builds on standard components, it introduces a boundary-aware, realism-constrained exploration of the generative distribution, which is not captured by existing adversarial robustness or realism-driven GAN generation frameworks. We will update Line 161 to clarify these distinctions.
>
> >**Q2-1: Generalization Mechanism from Single-Generator Exploration. (1) Shared artifact characteristics.**
>
> PROBE does not aim to cover the full distribution of unseen generators. Instead, it improves the detector’s decision boundary by exposing regions of uncertainty under realism constraints, which enhances its ability to generalize to unseen generators.
>
> We hypothesize that different generators produce images with common perceptual ambiguities (e.g., texture inconsistency, oversmoothing, semantic mismatch). By steering generation toward uncertain regions while maintaining realism, PROBE encourages the detector to rely less on generator-specific cues and more on transferable features.
>
> We provide two analyses. First, we project DALL·E 3 and Midjourney samples from Synthbuster, FLUX samples from AIGI-Quality-Paradox, and PROBE boundary-induced samples into the feature space of a ResNet50 detector, visualized via t-SNE. We observe substantial overlap between PROBE samples and those from unseen generators, indicating shared detector-relevant hard regions.
>
> Second, following [a], we extract noise patterns from generated images using a denoising network and analyze their energy spectra. PROBE samples and those from unseen generators exhibit similar spectral patterns.
>
> [a] Intriguing properties of synthetic images: from generative adversarial networks to diffusion models, CVPRW2023.
>
> >**Q2-2: (2) A failure case analysis.**
>
> We identify BigGAN as a failure case for ResNet50 detector: its performance remains low after PROBE. BigGAN forms an isolated cluster in the detector's feature space with minimal overlap with PROBE samples, and its frequency spectrum exhibits a distinct grid-like structure absent in other unseen generators. These results suggest that the gain is limited when an unseen generator lies outside these explored regions.
>
> Moreover, PROBE-DINOv2 achieves 95% accuracy on the BigGAN subset of GenImage, suggesting that stronger backbones with richer pretrained representations can help alleviate this limitation.
>
> >**Q3: Weak Justification for Not Performing Iterative Boundary Exploration.**
>
> We conducted iterative probing experiments on the ResNet50 detector, with average bAcc across seven benchmarks improving from 63.5 (baseline) to 78.1, 79.4, and 80.1 after 3 successive iterations. For reference, the competing method, DRCT, achieves 70.9. These results show that iterative probing can provide additional gains, but the marginal improvement decreases after the first round.
>
> ||ResNet50 baseline|iteration1|iteration2|iteration3|
> -|-|-|-|-|
> |Avg bAcc.|63.5|78.1|79.4|**80.1**|
> |Improvement over the previous iteration|-|**+14.6**|+1.3|+0.7|
>
> We also performed t-SNE visualization of samples across iterations and found that later iterations largely overlap with earlier ones, suggesting they revisit similar variation patterns with diminishing new coverage.
>
> We will revise Line 207 to state that one probing iteration captures most of the performance gain and offers a practical trade-off between cost and performance.
>
> >**Q4: Presentation of Table 2.**
>
> Thanks. We add a brief discussion of the results. The added text will be placed on L320, second column, as follows:
>
> _“As shown in Table 2, PROBE-DINOv2 consistently achieves strong AIGI detection performance across both GenImage and AIGI-Quality-Paradox, with average bAcc improvements of 5.1% and 1.0% over the second-best method, respectively.”_

---

> > ### Author Rebuttal · Reviewer_m4xY · 2026-04-02
> >
> > Thank you for the detailed and thoughtful response. My concerns regarding W2–W4 have been fully addressed.
> >
> > I acknowledge that this work provides a certain level of contribution. However, I still have reservations about its degree of novelty, as it largely builds upon existing methodologies. In my view, this does not sufficiently meet the novelty bar expected for a highly competitive venue such as ICML.
> >
> > Therefore, I will maintain my current score.

---

> > > ### Author Response · Authors · 2026-04-02
> > >
> > > Dear Reviewer m4xY,
> > >
> > > Thank you for your active engagement and for acknowledging that W2–W4 have been fully resolved. We respectfully disagree with the assessment regarding novelty and would like to clarify our perspective as follows:
> > >
> > > We openly acknowledge that PROBE uses widely-used components, including LoRA, perceptual regularization, and reward tuning. But this is precisely what these components were proposed for: to be adopted and combined by future researchers to solve new problems.
> > >
> > > What PROBE contributes is a ***new perspective: reframing AIGC detector generalization as closed-loop, boundary-aware exploration of the generative distribution under realism constraints***. To achieve this, to our knowledge, this is an early work that combines these modules in a simple and effective way to steer a diffusion model toward boundary-proximal regions in the generative space. Concretely, PROBE uses detector-guided diffusion to generate realistic boundary samples and feeds them back to refine the detector, shifting from passive sampling to active, detector-guided exploration.
> > >
> > > All reviewers recognize that this simple formulation leads to ***strong empirical performance***. As noted in your review, PROBE "_presents a sound methodological shift from passive data collection to active boundary exploration._" Reviewer **gK8x** highlights that the work identifies "_a key limitation of current AIGI detectors: insufficient coverage of the generative space,_" and that using the detector as a critic is "_intuitive and novel; a refreshing and meaningful angle._" Reviewer **YTGy** also notes that the work "_delivers a strong empirical contribution through comprehensive evaluation._" Reviewer **HUVf** further points out that PROBE "_reframes training data construction .. as active search in the generator manifold .. guided by the detector decision boundary, moving beyond standard sampling  and pixel-space adversarial examples._"
> > >
> > > Importantly, ***this type of contribution aligns with ICML’s definition of originality***, which "_does not necessarily require introducing an entirely new method_", but also equally values “_creative combinations of existing ideas_” and “_new perspectives that advance understanding._” ***PROBE provides such a perspective by connecting generator-side exploration with detector behavior in a closed loop, leading to improved generalization across unseen generators***.
> > >
> > > We hope this clarifies the conceptual contribution of PROBE and respectfully ask the reviewer to reconsider the assessment of novelty and adjust the score accordingly. We remain actively engaged and welcome any further discussion.
> > >
> > > Best,
> > >
> > > The Authors

---

### Official Review · Reviewer_gK8x · 2026-03-12

**Soundness:** 3
**Presentation:** 3
**Significance:** 3
**Originality:** 4
**Overall Recommendation:** 5
**Confidence:** 3

**Summary:**

This paper addresses the poor generalisation of AI-generated image (AIGI) detectors to images produced by unseen generators, arguing that the main cause is insufficient coverage of the generative space during training. The authors propose PROBE, a framework that improves detector robustness by actively exploring challenging regions of the generative space using a detector-guided diffusion generator. Specifically, the detector acts as a critic to guide the generator (via LoRA-based optimisation) to produce boundary-induced fake samples that are likely to be misclassified as real, while perceptual regularisation ensures visual realism. These challenging samples are then used to fine-tune the detector, encouraging it to learn more robust and generalizable features. Experiments across seven benchmarks show that PROBE consistently improves balanced accuracy and generalisation to unseen generators compared to existing AIGI detection methods.

**Compliance With Llm Reviewing Policy:**

Affirmed.

**Final Justification:**

The authors have resolved almost all of my concerns, so I keep my score.

**Key Questions For Authors:**

1. How sensitive is PROBE to the choice of the generator used for boundary exploration? Would probing with different generators produce significantly different samples?
2. What is the additional training time introduced by the probing stage compared to standard detector training?
3. The paper states that a single round of probing is sufficient. Did the authors experiment with multiple probing iterations, and if so, what were the results?

**Limitations:**

Yes

**Strengths And Weaknesses:**

**Strengths:**

1. The paper clearly identifies a key limitation of current AIGI detectors: insufficient coverage of the generative space during training, which provides a convincing explanation for the poor cross-generator generalisation observed in prior work.
2. The core idea of using the detector as a critic to probe the generative space is both intuitive and novel. This perspective connects detector training with generator-side exploration, which is a refreshing and meaningful angle.
3. The method is effective yet lightweight.

**Weaknesses:**

1. The method assumes access to a modifiable diffusion generator, which can be fine-tuned using LoRA. However, in many realistic settings, the generator might be closed-source, so the training access is unavailable, which might limit the applicability of the method in practical scenarios.
2. Although the method improves cross-generator generalisation, the probing process itself is conducted using a single simple diffusion model (e.g., Stable Diffusion 1.4 or 2.1).

---

> ### Author Rebuttal · Authors · 2026-03-31
>
> >**Q1: Impact of Inaccessible Generators on Model Fine-tuning in Real-World Scenarios.**
>
> **A:** Thank you for raising this important point. In fully black-box settings, a natural extension would be to replace generator fine-tuning with black-box optimization over prompts or other exposed generation controls [a], guided by detector feedback to search for hard samples. We view this as a promising direction for extending the framework. We will also incorporate this discussion into our paper as follows:
>
> _“PROBE is currently instantiated with a modifiable generator to enable direct exploration of challenging generative variations during detector refinement. Extending the framework to more restricted settings, such as fully black-box generators, may also be possible in principle by replacing generator fine-tuning with black-box optimization over prompts or other exposed generation controls, guided by detector feedback to search for hard samples.”_
>
> [a] DiffZOO: A Purely Query-Based Black-Box Attack for Red-teaming Text-to-Image Generative Model via Zeroth Order Optimization. Findings of NAACL 2025
>
> >**Q2-1: Impact of Generators Selection in PROBE. How sensitive is PROBE to the choice of the generator used for boundary exploration?**
>
> **A:** Thank you for this question. To investigate the sensitivity of PROBE to the choice of generator, we conduct experiments under both the ResNet50 and DINOv2 setups using SD 3.5-Medium, a more recent generator based on a fundamentally different architecture (DiT) from the U-Net-based SD 1.4/SD 2.1 used in our main experiments. The results are summarized in the table below (average bAcc across seven benchmarks), where we also report the performance of the second-best method in each setup for reference.
>
> |ResNet50| Basline|SD1.4|SD2.1|SD3.5-M|DRCT|
> |---|---|---|---|---|---|
> |Avg bAcc.|63.5|**78.1**|77.8|76.2|70.9|
>
> |DINOv2 |Basline|SD2.1|SD3.5-M|BFree|
> |---|---|---|---|---|
> |Avg bAcc.|87.4|93.9|**94.5**|89.3|
>
> We observe that PROBE remains stable with SD 3.5-Medium and continues to outperform the competing method in both setups. This suggests that PROBE is not tightly tied to a specific generator architecture, and that its benefit comes from the ability to explore challenging generative variations rather than from particular properties of the original probing model.
>
> >**Q2-2: Would probing with different generators produce significantly different samples?**
>
> **A:** The training cost for each stage is summarized below (all in GPU-hours on 2× NVIDIA A800-40GB).
>
> |Stage|ResNet50 (SD1.4)|DINOv2 (SD2.1)|
> |---|---|---|
> Baseline pretraining|3 h|10 h|
> PROBE: boundary exploration + sample generation|25 h|30 h|
> PROBE: detector fine-tuning|6 h|28 h|
>
> PROBE introduces additional cost beyond standard pretraining, mainly from the probing and sample generation stage. However, this overhead is moderate in practice for three reasons.
>
> First, we reduce cost through efficient design choices, including DRTune for lightweight adaptation and a single-round probing strategy, which already captures most of the performance gain. In practice, the cost is dominated by image generation rather than optimization. For ResNet50, generating 20k boundary-induced samples with PROBE takes about 25 GPU-hours, while standard SD 1.4 sampling of the same number of images already requires around 20 GPU-hours. This shows that the additional overhead from optimization is relatively small.
>
> Second, PROBE is significantly more data-efficient than comparable methods. DRCT and BFree require 236k and 258k generated samples, respectively, whereas PROBE achieves stronger generalization with only 20k samples:
> ||PROBE|DRCT|Bfree|
> |---|---|---|---|
> Generated samples number|20k|236k|258k|
>
> Finally, the cost of PROBE is a one-time offline process. At inference time, the detector architecture remains unchanged and introduces no additional latency.
>
> >**Q3: Quantitative Results of multiple probing iterations.**
>
> **A:** Thank you for this question. We did experiments with multiple probing iterations on the **ResNet50** detector. The average bAcc across seven benchmarks improves from **63.5** for the baseline to **78.1** after the first probing iteration, and further to **79.4** and **80.1** after the second and third iterations, respectively. For reference, the strongest competing method in our comparison, DRCT, achieves an average bAcc of **70.9**.
> |ResNet50|baseline|iteration1|iteration2|iteration3|DRCT|
> |-|-|-|-|-|-|
> |Avg bAcc.|63.5|78.1|79.4|**80.1**|70.9|
>
> These results suggest that iterative probing can bring additional gains. Since the first probing iteration already yields a clear improvement over prior methods, we use one probing iteration in the paper as a practical trade-off between computation and performance. We will also include full results for additional probing iterations in the supplementary material.

---

> > ### Author Rebuttal · Reviewer_gK8x · 2026-04-03
> >
> > The authors addressed all my concerns.

---

> > > ### Author Response · Authors · 2026-04-04
> > >
> > > Dear Reviewer gK8x,
> > >
> > > Thank you for your positive feedback and for maintaining a positive score. We appreciate your insightful suggestions and will incorporate them into the revision.
> > >
> > > Kind regards,
> > >
> > > The Authors

---

### Decision · Program_Chairs · 2026-04-30

**Decision:**

Accept (regular)

**Comment:**

This paper explores the generalization failure of AI-generated image detectors to unseen generators. The paper argues that standard sampling covers only a narrow portion of the generative space and proposes PROBE, a two-stage framework that uses a pre-trained detector as critic to steer a seen diffusion model toward its decision boundary.

Strengths.
+ The paper clearly identifies a key limitation of current AIGI detectors
+ The paper covers a variety of datasets and analysis

Weaknesses
- Limited novelty: The paper combines multiple existing components, which make technical novelty limited (m4xY, YTGy)
- Insufficient explanation for generalization. Generalization mechanism from a single generator exploration is unclear (m4xY, YTGy, HUVf)
- The paper claims to improve cross-generator generalization but experiments are limited to a single generator (gK8x)

The paper received four reviews with ratings 5,4,3,5 (confidence for reviewers is 3-4)
Authors submitted rebuttal and reviewers engaged in the discussion.

While most of the clarification questions were addressed, the concerns about novelty and generalization remain unresolved.

AC is primarily concerned about the lack of evidence on generalization. For instance, YTGy argued that t-SNE visualization showing overlap between PROBE samples and unseen generator samples does not constitute a rigorous justification for generalization.

Overall, a borderline paper with good empirical results with some outstanding concerns about the core claims.